# Nucleoporin107 mediates female sexual differentiation via Dsx

**Tikva Shore[1†], Tgst Levi[1†], Rachel Kalifa[1], Amatzia Dreifuss[1], Dina Rekler[1], Ariella Weinberg-Shukron[2], Yuval Nevo[3], Tzofia Bialistoky[1], Victoria Moyal[1], Merav Yaffa Gold[1], Shira Leebhoff[1], David Zangen[4], Girish Deshpande[5]*, Offer Gerlitz[1]***

[1]Department of Developmental Biology and Cancer Research, Institute of Medical Research Israel-Canada, The Hebrew University- Faculty of Medicine, Jerusalem, Israel; [2]Medical Genetics Institute, Shaare Zedek Medical Center, the Hebrew University Hadassah Medical School, Jerusalem, Israel; [3]Bioinformatics Unit of the I-CORE Computation Center, The Hadassah Hebrew University Medical Center, Jerusalem, Israel; [4]Division of Pediatric Endocrinology, Hadassah Hebrew University Medical Center, Jerusalem, Israel; [5]Department of Molecular Biology, Princeton University, Princeton, United States

**Abstract** We recently identified a missense mutation in Nucleoporin107 (Nup107; D447N) underlying XX-ovarian-dysgenesis, a rare disorder characterized by underdeveloped and dysfunctional ovaries. Modeling of the human mutation in *Drosophila* or specific knockdown of Nup107 in the gonadal soma resulted in ovarian-dysgenesis-like phenotypes. Transcriptomic analysis identified the somatic sex-determination gene *doublesex (dsx)* as a target of Nup107. Establishing Dsx as a primary relevant target of Nup107, either loss or gain of Dsx in the gonadal soma is sufficient to mimic or rescue the phenotypes induced by *Nup107* loss. Importantly, the aberrant phenotypes induced by compromising either *Nup107* or *dsx* are reminiscent of bone morphogenetic protein (BMP signaling hyperactivation). Remarkably, in this context, the metalloprotease AdamTS-A, a transcriptional target of both Dsx and Nup107, is necessary for the calibration of BMP signaling. As modulation of BMP signaling is a conserved critical determinant of soma–germline interaction, the sex- and tissue-specific deployment of Dsx-F by Nup107 seems crucial for the maintenance of the homeostatic balance between the germ cells and somatic gonadal cells.

*For correspondence:
gdeshpande@princeton.edu
(GD);
offerg@ekmd.huji.ac.il (OG)

†These authors contributed
equally to this work

**Competing interest:** The authors
declare that no competing
interests exist.

**Reviewing Editor:** Xin Chen,
Johns Hopkins University, United
States

## Editor's evaluation

This study uses *Drosophila* as a model to study a specific mutant in a gene encoding a nuclear pore protein, whose counterpart in human leads to a rare disease called XX-ovarian-dysgenesis. Intriguingly, the fly mutants mimic the syndromes identified in human patients, such as failures in ovary development and function. The authors use fly ovary as a model to study the reasons underlying the phenotypes, which should provide insight to our understanding of this disease.

## Introduction

Germline–soma communication lies at the heart of proper gonad development, and thus is essential for formation and coalescence of the primitive embryonic gonad until generation of the adult gonad (*Van Doren, 2006*; *Staab et al., 1996*; *Wawersik et al., 2005*). Gonadogenesis also relies upon coordination between cell autonomous and nonautonomous mechanisms that direct correct specification, patterning, and subsequent morphogenesis (*DeFalco et al., 2008*; *Nöthiger et al., 1989*).

Furthermore, the developmental program culminating in proper gonad formation must integrate both nonsex-specific 'housekeeping' functions and sex-specific signals to establish and maintain sexually dimorphic traits. Therefore, functional aberrations that affect individual molecular components, either sex- or nonsex-specific, result in various clinical disorders and infertility.

XX-ovarian-dysgenesis (XX-OD) is a rare, genetically heterogeneous disorder that is characterized by underdeveloped, dysfunctional ovaries, with a subsequent lack of spontaneous pubertal development, primary amenorrhea, uterine hypoplasia, and hypergonadotropic hypogonadism (*Hughes, 2008*). We recently identified a recessive missense mutation in the nucleoporin107 (*Nup107*) gene (c.1339G > A, p.D447N) as the causative mutation for isolated XX-OD (without other developmental deficits) in five female cousins from a consanguineous family (*Weinberg-Shukron et al., 2015*). All men in the family had normal pubertal development and those married have multiple children, indicating a female-specific nature of the functional consequences induced by the specific point mutation.

Nup107 is an essential component of the nuclear pore complex, enabling both active and passive transport in every nucleated cell (*Walther et al., 2003*). In light of this ubiquitous and vital function, the sex-specific and tissue-restricted nature of the *Nup107* XX-OD phenotype is highly intriguing. Of note, Nup107 and other NPC proteins were recently shown to have nuclear transport-independent activities including genome organization and tissue-specific regulation of gene expression (*Guglielmi et al., 2020*; *Gomar-Alba and Mendoza, 2019*; *Kuhn and Capelson, 2019*; *Pascual-Garcia and Capelson, 2021*; *Raices and D'Angelo, 2017*; *D'Angelo, 2018*; *Gozalo et al., 2020*). Taking into account the high conservation of the aspartic acid residue altered by the mutation, we used a *Drosophila* model to assess Nup107 function in human female gonadal development. Our previous studies demonstrated that RNAi-mediated knockdown (KD) of *Nup107* in somatic gonadal cells led to

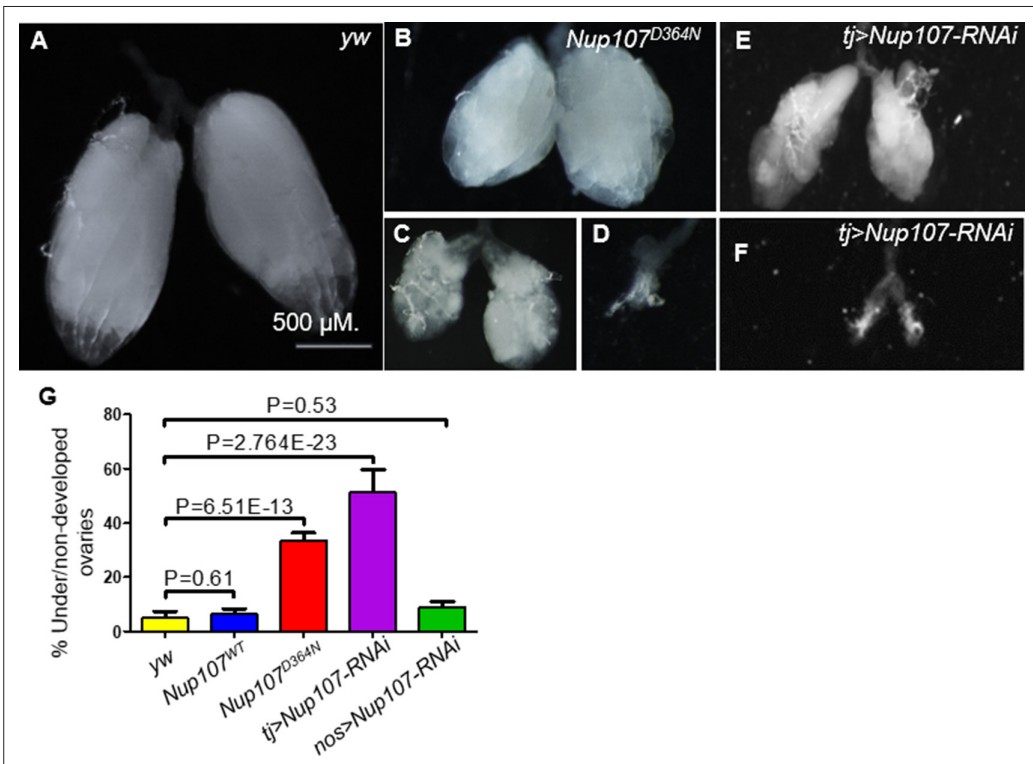

**Figure 1.** Phenotypic characterization of the ovaries compromised for *Nup107*. (**A**) *yw* control *Drosophila* ovaries. In contrast, *Nup107^D364N^* ovarian samples display a variety of aberrations, including (**B**) small or (**C**) shriveled ovaries, and (**D**) bilateral dysgenesis. Knockdown of *Nup107* using *tj-Gal4* driver recapitulated the same phenotypes as *Nup107^D364N^*, with flies exhibiting (**E**) underdeveloped or (**F**) nondeveloped ovaries. (**G**) The percentages of under/nondeveloped ovaries in *yw* (n = 196), *Nup107^WT^* (n = 504), *Nup107^D364N^* (n = 890), *tj > Nup107* (n = 316), and *nos > Nup107* (n = 64) flies. Scale bar in A applies to all panels.

The online version of this article includes the following figure supplement(s) for figure 1:

**Figure supplement 1.** Gene knockdowns were validated through quantitative real-time PCR (qRT-PCR).

female-specific sterility due to defective oogenesis, while male flies developed normally and remained fertile (*Weinberg-Shukron et al., 2015*). Furthermore, generating a *Drosophila* model of the human mutation by introducing an *RFP-Nup107^WT^* transgene in *Nup107*-null flies rescued both lethality and fertility of female flies. Intriguingly, however, introduction of an *RFP-Nup107^D364N^* transgene essentially recapitulated the familial XX-OD mutation. While it rescued lethality, the mutation resulted in severely reduced female fertility, ovariole disintegration, and extensive apoptosis (*Weinberg-Shukron et al., 2015*).

A second mutation in *Nup107* (c.1063C > T, p.R355C) has since been independently identified as a cause of XX-OD (*Ren et al., 2018*). Notably, our analysis had predicted salt bridge interactions between the altered Aspartic Acid 447 in our study and the Arginine 355 altered in the second family, strengthening the notion that Nup107 performs an essential and conserved ovary-specific function during female gonadogenesis. Here, to uncover the molecular underpinnings of Nup107's function, we analyzed the 'loss of function' phenotypes of *Nup107* at both the cellular and transcriptional levels.

## Results

### *Nup107* mutant ovaries display an ovarian-dysgenesis-like phenotype

A closer inspection of *Nup107^D364N^* adult ovaries revealed that the female gonads displayed phenotypic traits closely resembling human ovarian dysgenesis. Specifically, the phenotypes ranged from rudimentary, small ovaries with fewer ovarioles to bilateral dysgenesis where both ovaries were completely absent (*Figure 1B–D*). Approximately 33% of *Nup107^D364N^* ovaries were either under- or nondeveloped, in comparison to milder defects in only 6% and 5% in *Nup107^WT^* and *yw* control flies, respectively (*Figure 1A, G*). We found similar ovarian defects when inserting the *Nup107* mutation (1090G > A, p.D447N) into the *Drosophila* genome using CRISPR (*Levi et al., 2020*).

To discern if Nup107 is necessary in the somatic or germline component of the gonad, or in both, we selectively inactivated *Nup107* function using *UAS-Nup107-RNAi* in combination with either a somatic gonadal specific driver *traffic jam–Gal4* (*tj-Gal4*) or a germ cell-specific driver *nanos-Gal4* (*nos-Gal4*). Interestingly, only soma-specific KD of *Nup107* recapitulated the mutant phenotype, where 51% of adult ovaries were 'dysgenic' or underdeveloped (*Figure 1E–G*). Supporting the conclusion that in this functional context Nup107 is required primarily in the soma, germline-specific inactivation of *Nup107* did not lead to significant phenotypic abnormalities compared to the control (9%, *Figure 1G*) (see *Figure 1—figure supplement 1* for validation of *RNAi* KD).

### Soma–germline homeostasis in the larval gonad requires Nup107

To trace back the phenotypic consequences of Nup107 loss to the earlier stages of gonad development, we sought to analyze the third instar larval stage (LL3) gonads. At this stage, the *Drosophila* larval gonad consists of a somatically derived stem cell niche and primordial germ cells (PGCs) adjacent to it (*Figure 2A*). Interspersed among the PGCs are their somatic support cells, known as intermingled cells (ICs). Dissection of *yw*, *Nup107^WT^*, and *Nup107^D364N^* larvae showed that their gonads were equally present and readily identifiable in all three genotypes (*Figure 2—figure supplement 1A*).

To assess if the somatic and germline components of the larval gonad are specified and patterned correctly in a *Nup107* compromised background, we stained larval gonads from both *Nup107^D364N^* flies and those with somatic KD of *Nup107* (hereafter *tj > Nup107* KD), using antibodies which specifically mark the PGCs (VASA) and ICs (TJ). Confocal imaging of these larval gonads revealed that both cell types are present at this developmental stage, however their spatial organization appeared to be disrupted. Moreover, both *Nup107^D364N^* and *tj > Nup107* KD larval gonads contained excess numbers of PGCs (*Figure 2C–E*) which formed large clusters devoid of ICs (*Figure 2E'1, J*). While the total number of mutant ICs was unaffected (*Figure 2—figure supplement 1B*), these cells failed to intermingle with the PGCs and often remained clustered at the periphery of the gonadal tissue (*Figure 2E–H*). Staining with antibodies against an adducin-like molecule (1B1) which marks the fusome, a subcellular organelle, revealed elevated numbers of cells with spherical fusomes, normally characteristic of germline stem cells (GSCs, *Figure 2—figure supplement 1C–E*). Proper soma–germline communication is necessary for the maintenance of the homeostatic balance required for adequate proliferation and differentiation of both cell types. Altogether these data show that loss

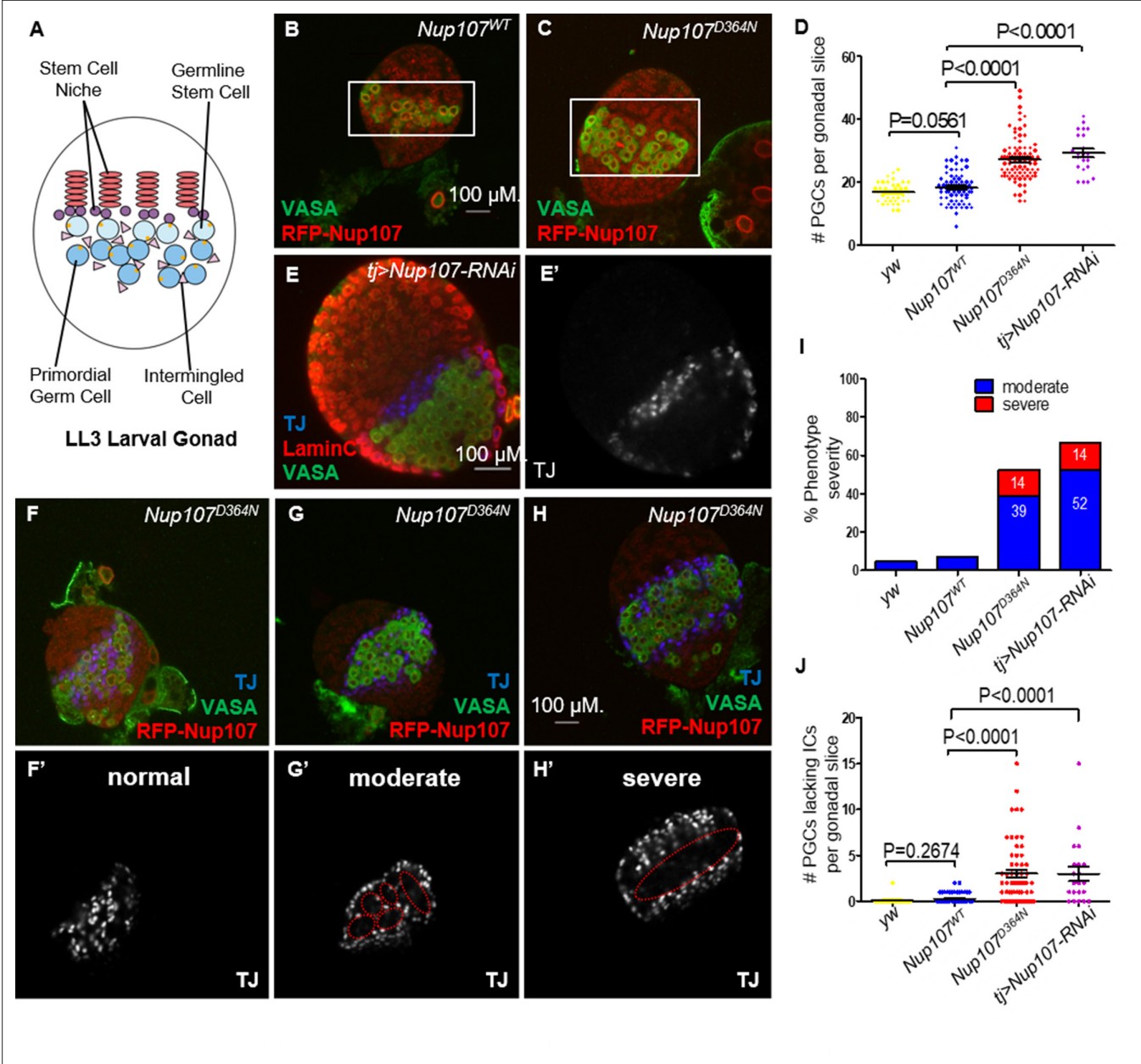

**Figure 2.** *Nup107^{D364N}* larval gonads display aberrant cellular number and arrangement. (**A**) Schematic representation of the LL3 larval gonad with its different cell types. (**B**) Confocal section of *Nup107^{WT}* gonad containing on average 18 primordial germ cells (PGCs; VASA, green), compared to (**C**) 27 in *Nup107^{D364N}* gonads. (**D**) Quantitation of the total PGCs per confocal section in each gonad, in *yw*, *Nup107^{WT}*, *Nup107^{D364N}*, and *Nup107* KD (*n* = 42, 71, 88, 22). (**E**) *Nup107* KD larval gonads contain excess PGCs (VASA, green) and abnormally dispersed intermingled cells (ICs; TJ, blue). *Nup107^{D364N}* larval gonads exhibit a range of IC dispersion patterns from (**F**) normal to (**G**) moderate to (**H**) severe. (**I**) The percentage of each IC phenotype found in *yw*, *Nup107^{WT}*, *Nup107^{D364N}*, and *Nup107* KD gonads (*n* = 22, 61, 58, 21). (**J**) Representation of the number of PGCs per gonad missing an immediately adjacent IC, due to abnormal dispersion.

The online version of this article includes the following figure supplement(s) for figure 2:

**Figure supplement 1.** Female larval gonads compromised for *Nup107* show characteristic aberrations.

of *Nup107* in the somatic gonadal precursors affects the soma–germline communication, adversely affecting the total PGC count and relative positioning of the ICs.

### *Nup107* mutant ovaries display increased bone morphogenetic protein signaling activity away from the GSCs niche

We previously found that *Nup107* mutant female flies with normal-sized ovaries suffered from infertility due to defective oogenesis. To understand the functional underpinnings of the adult ovarian phenotypes induced by the *D364N* mutation we sought to analyze the *Nup107* mutant ovaries carefully. The *Drosophila* ovary is made up of 16–20 ovarioles that function individually as egg production lines. The germarium, situated at the anterior tip of the ovariole, contains the somatic stem cell niche and adjacent GSCs (*King, 1970*; *Spradling, 1993*). Typically, a GSC divides asymmetrically to generate another stem cell and a cystoblast. Cystoblasts divide and differentiate to form 16 interconnected cystocytes, with the dynamically expanding fusome acting as the connecting link. Fusome morphology can be

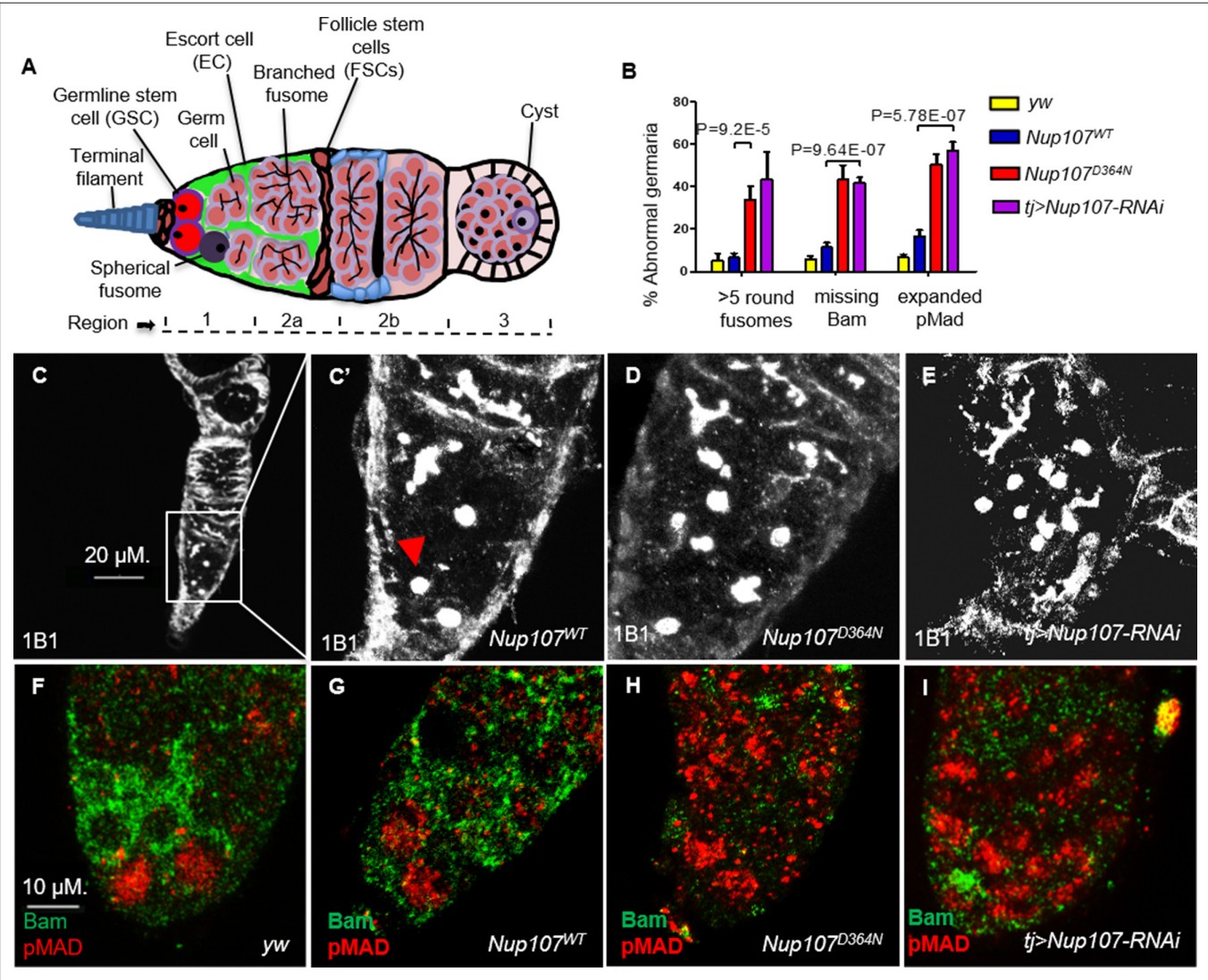

**Figure 3.** *Nup107^{D364N}* adult ovaries demonstrate increased germline stem cell (GSC) number. (**A**) A scheme of the *Drosophila* germarium. In region 1, GSCs divide four times to form a 16-cell cyst. Region 2a is identified by the presence of 16-cell cystocytes which are adjoining the follicle stem cell border. (**B**) Quantification of spherical fusomes (*n* = 57, 74, 100, 27), Bam (*n* = 127, 137, 139, 64), and pMad (*n* = 122, 121, 225, 48) phenotypes in *yw*, *Nup107^{WT}*, *Nup107^{D364N}*, and *Nup107* KD, respectively. (**C**) *Nup107^{WT}* germaria contain two to three cells with spherical fusomes, indicated by arrowhead, while (**D**) *Nup107^{D364N}* and (**E**) *tj > Nup107* germaria contain an average of six cells with spherical fusomes. (**F**) *yw* and (**G**) *Nup107^{WT}* germaria show normal pMad (blue) and Bam (green) expression compared to (**H**) *Nup107^{D364N}* and (**I**) *tj > Nup107* germaria which show reduced Bam levels and expanded pMad expression.

assessed by using antibodies against Huli-tai-shao, 1B1 (*de Cuevas and Spradling, 1998*, *Lin and Spradling, 1995*). Thus, GSCs and their undifferentiated cystoblasts can be readily identified by the presence of a round fusome, whereas their differentiated daughter cells display extended, branched fusomes (*Figure 3A*). Ovary staining revealed that unlike *yw* and *Nup107^WT^* germaria, which contained two to three spherical fusomes, *Nup107^D364N^* germaria contained an average of six spherical fusomes per germarium (7% of *Nup107^WT^* vs. 34% of *Nup107^D364N^*; *Figure 3B–D*). Similar results were seen in *tj > Nup107* KD (*Figure 3B, E*) germaria. Thus, *Nup107* compromised germaria contain elevated numbers of cells with spherical fusomes, characteristic of both GSCs and undifferentiated cystoblasts. Interestingly, an increase in the total number of undifferentiated germ cells is reminiscent of hyperactivation of the bone morphogenetic protein (BMP) signaling pathway (*Zhu and Xie, 2003*).

We thus wondered if inappropriately expanded BMP signaling is responsible for the failure in GSC differentiation. The somatic terminal filament (TF) and Cap cells (CCs), which together constitute the GSC niche, normally secrete a BMP ligand, Decapenataplegic (Dpp) (*Xie and Spradling, 1998*; *Xie and Spradling, 2000*). Binding of Dpp to its cognate receptor Thickveins (Tkv), expressed in GSCs, triggers a signal transduction cascade, which ultimately represses the expression of the master differentiation gene *bag of marbles* (*bam*) in the GSC region (*McKearin and Ohlstein, 1995*). As GSC daughter cells move away from the niche, BMP signaling is progressively weakened, resulting in the induction of Bam expression. We found that Bam expression was continuously repressed and completely absent in 43% of *Nup107^D364N^* and 42% of *tj > Nup107* KD regions 1 and 2a germaria, compared to 6% and 11% in *yw* and *Nup107^WT^* flies, respectively (*Figure 3B, F–I*). As phenotypic consequences resulting from *Nup107* loss seemed analogous to those induced by excess BMP signaling, we therefore sought to analyze the downstream components of the BMP pathway.

The transcriptional response to the BMP signal that emanates from the stem cell niche is mediated by phosphorylation of MAD (pMad) which translocates to the nucleus and with its binding partner, Medea, regulates pathway targets (*McKearin and Ohlstein, 1995*; *Feng and Derynck, 2005*; *Schmierer and Hill, 2007*). pMad represses *bam* and thus allows for maintenance of the undifferentiated state and self-renewal of GSCs (*Song et al., 2004*). We found that pMad levels in *Nup107* compromised germaria are abnormally elevated in regions distant from the GSC niche. Specifically, 50% of *Nup107^D364N^* and 57% of *Nup107* KD germaria show high expression of pMad, compared to 7% of *yw* and 17% of *Nup107^WT^* germaria (*Figure 3B, F–I*). Taken together, these findings support the notion that compromising Nup107 activity leads to hyperactivation of BMP/Dpp signaling away from the GSC niche and suggest that Nup107 restricts the range and/or strength of Dpp signaling required for proper differentiation of the GSCs.

## Impairment of the GSC progeny differentiation niche in *Nup107* mutant ovaries

Escort cells (ECs) and their cellular processes, which encapsulate the GSC daughter cells that leave the stem cell niche, together constitute a distinct niche responsible for controlling germ cell differentiation (*Kirilly et al., 2011*; *Wang et al., 2011*). This is thought to be achieved, in part, by restricting BMP signaling (*Kirilly et al., 2011*; *Wang et al., 2011*). ECs employ Hh, Wnt, EGFR, and Jak-Stat signaling to prevent BMP signaling in GSC progeny (*Wang and Page-McCaw, 2018*; *Tseng et al., 2018*; *Lu et al., 2015*; *Huang et al., 2017*; *Mottier-Pavie et al., 2016*; *Wang et al., 2015*; *Luo et al., 2015*; *Hamada-Kawaguchi et al., 2014*; *Upadhyay et al., 2016*; *Maimon et al., 2014*). Most, if not all of these signals, act to sustain the cytoskeletal structure of the processes that emerge from the ECs. The accumulation of undifferentiated GSCs and the expanded range of BMP signaling observed in *Nup107* compromised germaria could result from the failure of the EC differentiation niche function. We reasoned that such a failure may be reflected in the ability of the cellular extensions to emerge from the ECs. To test this idea, ovaries were costained with anti-Traffic Jam (TJ) and anti-Coracle (Cora). Cora is a structural protein that is highly expressed in the cellular extensions of ECs (*Maimon et al., 2014*; *Fairchild et al., 2015*) whereas TJ is a transcription factor expressed in the nuclei of ECs, CCs, and follicle stem cells in the germarium (*Li et al., 2003*; *Jin et al., 2013*; *Weaver et al., 2020*). The cellular extensions of the ECs in *yw* and *Nup107^WT^* germaria were readily detected (*Figure 4A, B*). In contrast, these cellular extensions were either dramatically reduced or completely lost in over 36% and 48% of the *Nup107^D364N^* and *tj > Nup107* KD germaria, respectively (*Figure 4C, D, G*). Furthermore, the reduction in cellular extensions was unlikely to be due to loss of ECs as they were

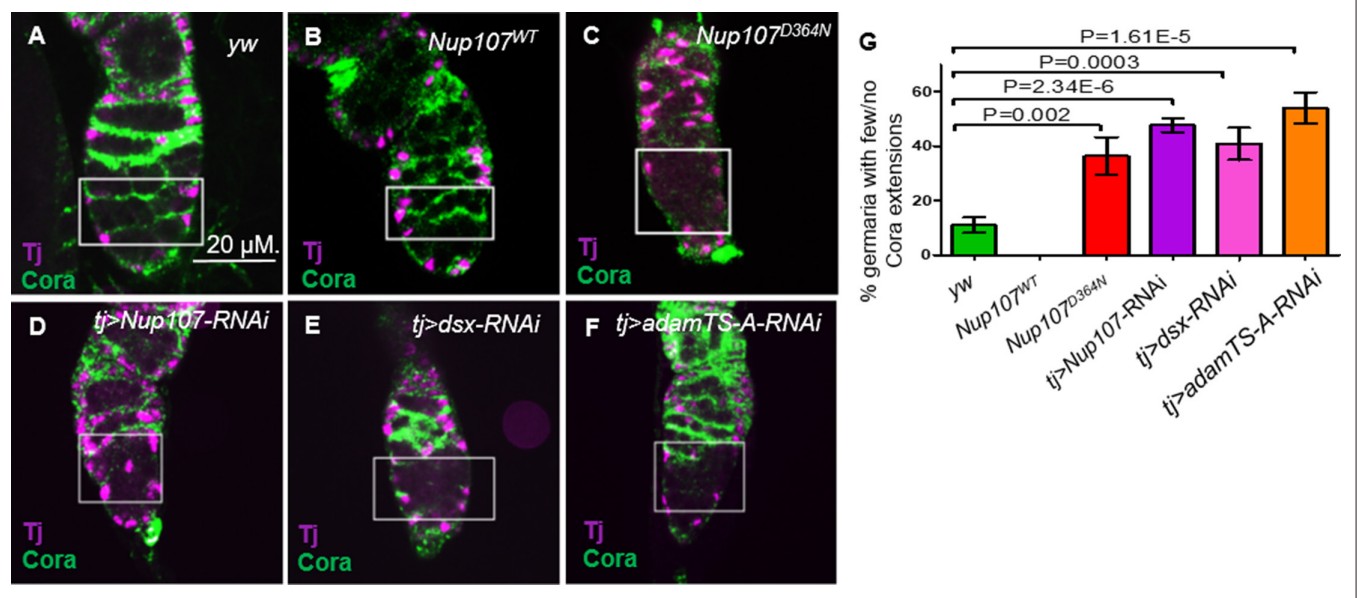

**Figure 4.** Adult ovaries demonstrate missing escort cell (EC) extensions. In germaria taken from (**A**) *yw* and (**B**) *Nup107^{WT}*, the ECs extensions were easy to note. In contrast, ECs extensions from (**C**) *Nup107^{D364N}*, (**D**) *tj > Nup107* KD, (**E**) *tj > dsx* KD, and (**F**) *tj > AdamTS-A* KD germaria were dramatically reduced or lost. (**G**) Quantitation of Coracle extensions from *yw* (n = 88), *Nup107^{WT}* (n = 32), *Nup107^{D364N}* (n = 25), *tj > Nup107* KD (n = 65), *tj > dsx* KD (n = 60), and *tj > AdamTS-A* KD (n = 54). Anti-Cora and anti-TJ staining are shown in green and magenta, respectively.

The online version of this article includes the following figure supplement(s) for figure 4:

**Figure supplement 1.** Knockdown of Nup107 in ECs results in loss of their cellular extensions accompanied with accumulation of undifferentiated GSCs.

**Figure supplement 2.** Knockdown of *cora* in adult ovaries results in fewer escort cell (EC) extensions without affecting EC number.

**Figure supplement 3.** *cora* KD in adult ovaries showed aberrant phenotypes similar to *Nup-107* KD.

readily visible, when labeled with anti-Traffic jam, in both WT and *Nup107* compromised germaria (*Figure 4A–D*).

To extend these data we used another driver, *c587-Gal4*. The expression pattern of *c587-Gal4* driver is similar to *tj-Gal4* in the adult germarial areas, except that it is not active in CCs and adult follicle cells (*Jin et al., 2013*; *Weaver et al., 2020*). KD of *Nup107* using the *c587-Gal4* driver also resulted in considerable reduction in the cellular extensions arising from the ECs, expanded pMad expression and accumulation of undifferentiated germ cells (*Figure 4—figure supplement 1*). Together these observations suggested that Nup107 activity is primarily required in the ECs. Notably, ECs were readily detected, as assessed by the presence of GFP-positive cells, in both WT and *Nup107* KD germaria. These data further indicated that in our experimental setup, viability of ECs is unaffected due to loss of Nup107.

To directly assess the role of the cellular extensions derived from the ECs in restricting BMP signaling, we sought ways to inhibit their formation without directly interfering with the intracellular signaling with in the ECs. To this end, we knocked down *coracle* in ECs using the *c587-Gal4*. Concomitant with the loss of the protrusions arising from the ECs (*Figure 4—figure supplement 2*), the resulting germaria exhibited accumulation of germ cells with round fusomes and expanded pMad expression away from the TF/CCs niche (*Figure 4—figure supplement 3*). These aberrant phenotypes were not due to an absence of ECs in the affected area, as similar number of ECs were visible (Green Fluorescent Protein (GFP)-positive cells), in the control as well as *coracle* compromised germaria (*Figure 4—figure supplement 2*). Together, these data strongly suggest that compromising *Nup107* impairs the formation of the extensions emanating from the ECs, which are required for restricting the BMP signal in the GSC differentiation niche.

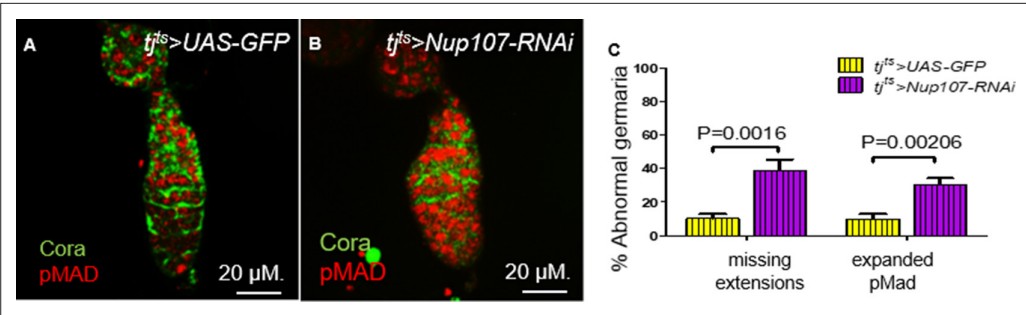

**Figure 5.** Knockdown of *Nup107* specifically at the adult stage resulted in loss of escort cell (EC) extensions. In germaria taken from (**A**) *tjt^s^ > GFP* flies, both the two to three germline stem cells (GSCs; pMad, red) and EC extensions (Cora, green) were easy to distinguish. (**B**) In contrast, pMad expression was expanded, and ECs extensions were absent in *tjt^s^ > Nup107* RNAi germaria. (**C**) Quantification of abnormal germaria with missing EC extensions or expanded pMad expression in *tjt^s^ > GFP* (n = 69) and *tjt^s^ > Nup107* RNAi (n = 74) flies. Bars represent mean + standard error of the mean (SEM). p values are from a two-tailed *t*-test.

## Nup107 activity is continuously required in ECs for the formation of cellular extensions and regulation of GSCs differentiation

Data described thus far showed that loss of Nup107 impairs the differentiation of the ECs. Consequently, ECs compromised for *Nup107* activity lost the cellular extensions possibly diminishing their ability to establish a productive communication with the GSCs in the vicinity and promote their differentiation. The adult ovarian ECs are the descendants of the larval ovarian ICs (*Reilein et al., 2021*). Collectively, our results are consistent with the model that Nup107 activity is required in the larval gonad for the specification of the ICs, whereas the adult ovarian phenotypes reflect a secondary consequence of compromising Nup107 activity in the EC progenitors. Our data, however, do not rule out the possibility that Nup107 activity is continuously required in the adult germarium for the proper functioning of the ECs. To distinguish between these two possibilities, we engineered a strategy involving a temporal KD of *Nup107* using a combination of *tj-Gal4* and a temperature sensitive form of the Gal4 repressor, *Gal80^ts^* (hereafter *tj^ts^*). We allowed for normal ovarian development until hatching and only then knocked down *Nup107* in adult somatic gonadal cells. Subsequent dissection of 4- to 5-day-old female flies showed that the resulting ovaries were similar in size to their WT counterparts. Staining for Cora, however, revealed loss of ECs' cellular extensions and expansion of pMad expression (*Figure 5*), similar to the phenotypes observed when *Nup107* was compromised continuously from early larval stages. Thus, we concluded that Nup107 activity enables the formation of cellular extensions arising from the ECs, which are essential for their proper function during GSC differentiation.

## Sex-determination gene *Dsx* is a target of Nup107

To better understand Nup107 function in the ovary, we sought to explore if the *Nup107 D364N* mutation influences transcription in the female larval gonads and if so, whether the possible changes in the transcriptional profile could be correlated with the developmental defects observed in *Nup107* mutant ovaries. To assess the genome-wide transcriptional changes in the *Nup107* mutant ovaries, we performed an unbiased transcriptome analysis on *yw*, *Nup107^WT^*, and *Nup107^D364N^* LL3 larval female gonads, in triplicates, using RNA-seq (GEO accession number GSE141094). We identified 82 candidate genes (*Supplementary files 1 and 2*) which displayed significant changes in mRNA expression in the larval gonad upon compromising *Nup107*. Among these candidates, we were particularly intrigued by the decreased expression of the DMRT transcription factor family member *doublesex* (*dsx*), which is critical for sex-specific differentiation (*Matson and Zarkower, 2012*; *Hildreth, 1965*; *Baker and Ridge, 1980*). We confirmed that *dsx* is a target of Nup107 using quantitative real-time PCR (qRT-PCR; (*Figure 6—figure supplement 1*).

*dsx* is expressed in both sexes, but is regulated via sex-specific alternative splicing, which results in the generation of either female (*dsx^F^*)- or male (*dsx^M^*)-specific isoforms (*Baker and Wolfner, 1988*; *Burtis and Baker, 1989*; *Burtis et al., 1991*). Supporting the conclusion that both Dsx^F^ and Dsx^M^ are determinants of proper sexual development, either loss of individual Dsx function or simultaneous

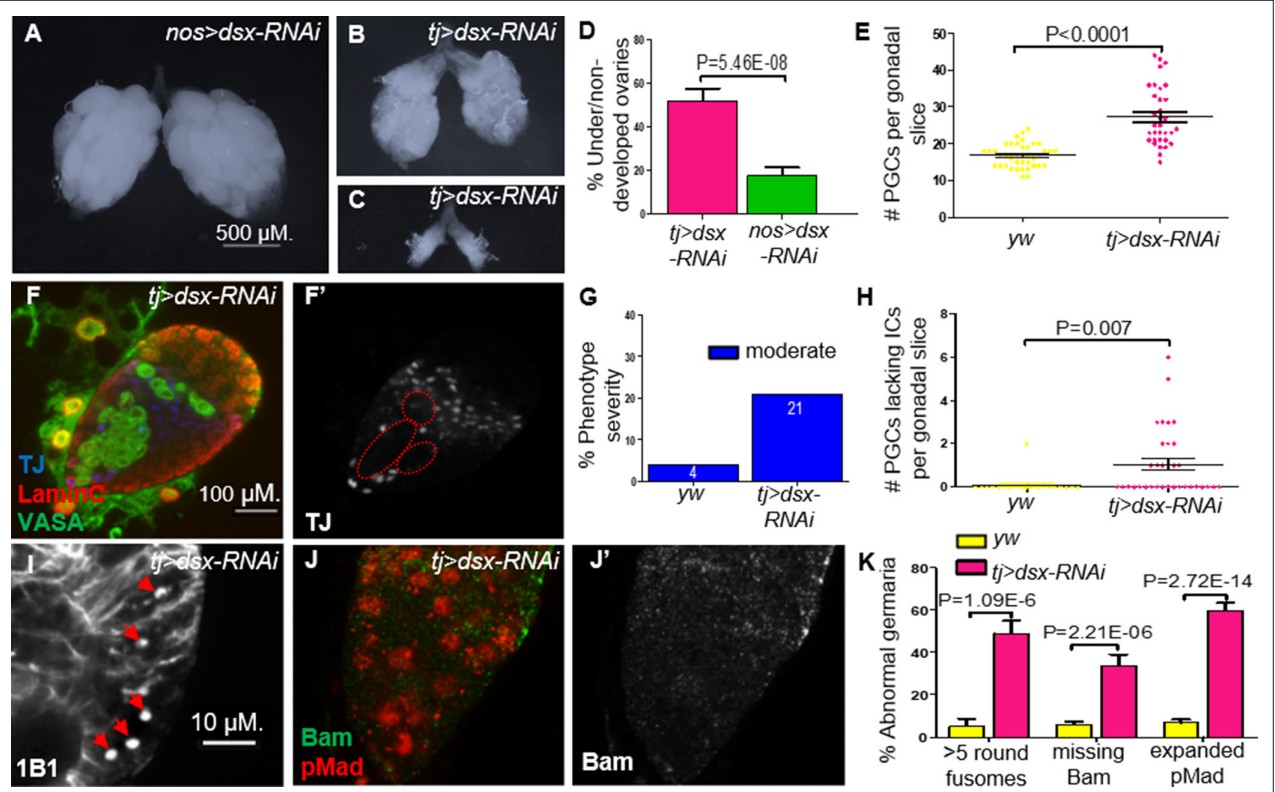

**Figure 6.** Knockdown (KD) of *dsx* in the gonadal soma recapitulates *Nup107^D364N* phenotypes. (**A**) *dsx* KD in the germline (*nos-Gal4*) results in negligible effects, compared to KD in somatic cells (*tj-Gal4*), which results in both (**B**) underdeveloped and (**C**) nondeveloped ovaries. (**D**) The percentages of under/nondeveloped ovaries in *tj-Gal4 dsx* KD (*n* = 74) vs. *nos-Gal4 dsx* KD (*n* = 100). (**E**) Representation of the number of primordial germ cells (PGCs) per confocal section of each individual gonad in *yw* (*n* = 42) and *dsx* KD (*n* = 34) larvae. (**F**) Larval gonads where *dsx* is KD using *tj-Gal4* contain excess PGCs and abnormally dispersed intermingled cells (ICs). (**G**) The percentage of IC severity phenotypes found in *yw* (*n* = 22) and *dsx* KD (*n* = 16) larval gonads. (**H**) Quantitation of the number of PGCs per gonad lacking an immediately adjacent IC, as a result of abnormal IC dispersion in *yw* (*n* = 22) and *dsx* KD (*n* = 34) larval gonads. (**I**) *tj*-driven *dsx* KD results in excess spherical fusomes, as well as (**J**) excess pMad (red) expression and missing Bam (green). (**K**) Quantitation of *dsx* KD germaria aberrant phenotypes (*n* = 53, 119, and 63, respectively).

The online version of this article includes the following figure supplement(s) for figure 6:

**Figure supplement 1.** Reduction in *dsx* and *AdamTS-A* transcription is observed upon compromising Nup107 activity in the *Drosophila* larval gonad.

gain of both *dsx^F* and *dsx^M* results in an intersexual phenotype (***Clough et al., 2014***). Interestingly, despite its crucial role in establishing and maintaining sexually dimorphic differentiation and behavior, *dsx* is expressed in only a select subset of tissues (***Sanders and Arbeitman, 2008***; ***Yang et al., 2008***). Establishing the biological relevance of the localized expression, inactivation of *dsx* using *dsx-Gal4* resulted in reduction in the size of the ovaries (***Clough et al., 2014***). As *dsx* expression was adversely influenced upon ovary specific loss of Nup107, we sought to test if Dsx plays an important role in female differentiation downstream of Nup107.

To assess this possibility, we first tested if compromising *dsx* in the somatic component of the gonad can mimic the *Nup107^D364N* phenotypes. Indeed, KD of *dsx* expression using somatic driver *tj-Gal4*, resulted in ovarian defects including either partial or complete dysgenesis (in 52% of adult ovaries; ***Figure 6A–D***). Importantly, compromising *dsx* using *nos-Gal4*, a germline-specific driver, did not yield similar phenotypes. Furthermore, *dsx* KD using a *tj-Gal4* driver also led to excess numbers of PGCs as well as a significant number of larval gonads with aberrant IC distribution (***Figure 6E–H***). Analysis of germaria compromised for Dsx^F function in the ovarian soma largely mimicked the loss of *Nup107* with respect to excess GSCs (distinguishable by the elevated number of spherical fusomes; ***Figure 6I, K***) and loss of the cellular extensions of the ECs (***Figure 4E, G***). Importantly, as in the case of *Nup107*, loss of *dsx^F* in the ovarian soma correlated with aberrant BMP signaling, as reflected in the

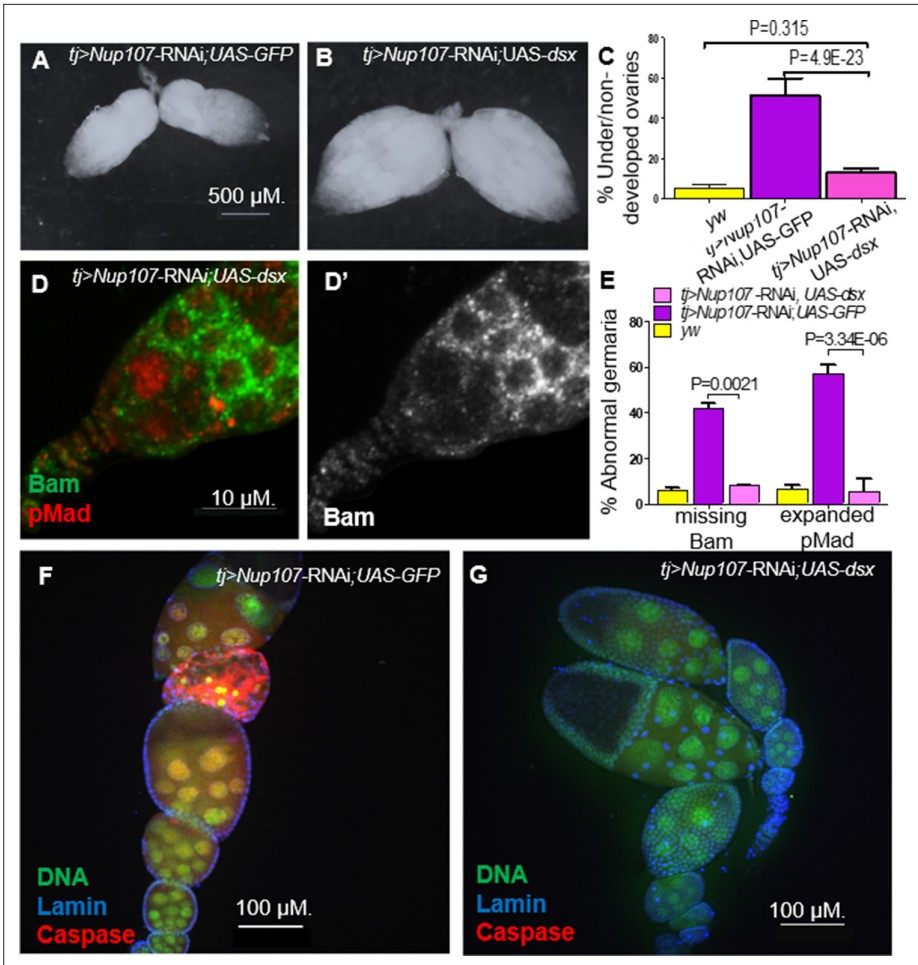

**Figure 7.** Overexpression of *Dsx* rescues *Nup107* KD ovarian phenotypes. (**A**) *RNAi*-KD of *Nup107* using *tj-Gal4* driver results in small, underdeveloped ovaries. (**B**) Coexpression of *dsx* with *RNAi*-KD of *Nup107* rescues the underdeveloped phenotype, resulting in normal, robust ovaries. (**C**) Quantitation of under/nondeveloped ovaries in *yw* (196), *Nup107-KD; UAS-GFP* (n = 316) and *Nup107* KD; *dsx* overexpression flies (n = 250). (**D**) *tj-Gal4*-driven *Nup107* KD, *dsx* overexpression germarium contains normal pMad (red) expression and (**D'**) normal Bam expression. (**E**) Quantitation of Bam and pMad expression in *yw* (n = 127, 122), *Nup107* KD; *UAS-GFP* (n = 64, 48), and *Nup107* KD; *dsx* overexpression (n = 25, 33) germaria. (**F**) 55% of *tj-Gal4*-driven *Nup107* KD; *UAS-GFP* ovarioles (n = 20) showed apoptosis, marked by anti-Caspase3 (red) compared to (**G**) zero *Nup107* KD, *dsx* overexpression ovarioles (n = 15).

accumulation of excess nuclear pMad accompanied by the loss of Bam in region 2a of the germaria (***Figure 6J, K***).

### *Dsx* overexpression rescues the phenotypes induced by *Nup107* loss

The similarity between the *Nup107* mutant and *dsx* KD ovarian phenotypes prompted us to test whether Dsx is a critical target of Nup107. If so, overexpression of $dsx^F$ alone should mitigate the phenotypes induced by the loss of *Nup107*. We therefore knocked down *Nup107* and concomitantly overexpressed $dsx^F$, utilizing *GFP* overexpression as a control in the KD lines. Analysis of the resulting ovarian samples revealed that the proportion of underdeveloped ovaries was substantially diminished as compared to both $Nup107^{D364N}$- and *tj-Gal4*-driven *Nup107* KD flies (***Figure 7A–C***). The gonads rescued upon simultaneous overexpression of $dsx^F$ appeared morphologically normal (***Figure 7B***), resembling those of control *yw* females. Furthermore, normal patterns and levels of both pMad and Bam expression were restored in regions 1 and 2a of the germaria (***Figure 7D, E***). Importantly, immunostaining of apoptotic marker caspase revealed that the *dsx*-rescued ovaries no longer demonstrated any abnormal cell death (***Figure 7F, G***), as shown in our previous findings (***Weinberg-Shukron et al.,***

*2015*). Taken together these data strongly suggested that Dsx is a key component acting downstream of Nup107 and loss of Dsx$^F$ could account for the stage- and sex-specific phenotypes associated with compromised Nup107 activity.

The nearly complete rescue observed by the introduction of *dsx*$^F$ transgene also implied that the two determinants could share important targets. We tested the notion by comparing the 82 genes, identified as Nup107 targets, with previously reported Dsx$^F$ targets (*Clough et al., 2014*). Indeed, the comparison revealed that 47 out of the 82 Nup107 targets (57%, p < 4.5E−11) were also identified as targets of Dsx (*Supplementary file 2* ). In addition to several transcription factors, included in this list are known modulators of BMP signaling as well as multiple components of the extracellular matrix (ECM). The substantial overlap between the two lists thus provides the key to the nearly complete rescue. Importantly these data have begun to elucidate how germline–soma communication engineered by the Dsx$^F$ can contribute to the establishment of female germline identity.

## The extracellular metalloprotease AdamTS-A acts downstream of Nup107 and Dsx

Our data demonstrate that Dsx$^F$ regulates the range and/or the strength of BMP signaling from the niche. We were intrigued by AdamTS-A, another candidate we identified, a secreted metalloprotease required for normal germ cell development (*Ismat et al., 2013*) that controls ECM assembly. Interestingly, human ovarian disorders including polycystic ovary syndrome and primary ovarian insufficiency have been correlated with compromised AdamTS-A activity (*Ozler et al., 2017*; *Russell et al., 2015*; *Knauff et al., 2009*). Using qPCR analysis, we confirmed that *AdamTS-A* expression is appreciably reduced in adult female ovaries compromised for Nup107 activity as compared to the wild type (*Figure 8G*, *Figure 6—figure supplement 1*). We questioned if the downregulation of *AdamTS-A*, downstream of Nup107, is mediated by Dsx$^F$ as it was previously reported to be a target of Dsx (*Clough et al., 2014*). Supporting the conclusion that transcription of *AdamTS-A* is positively regulated by Dsx$^F$, we observed a significant decrease in the levels of *AdamTS-A* transcripts following KD of *dsx* (*Figure 8G*).

Further, we sought to test if downregulation of *AdamTS-A* is an important determinant of ovarian development downstream of Nup107 and/or Dsx$^F$. KD of *AdamTS-A* in the somatic gonadal cells had severe effects on ovarian development including partial ovarian dysgenesis (*Figure 8B, C*), defective distribution of ICs and increased number of PGCs (*Figure 8D–F*) and GSCs (*Figure 8H1*) in the larval and adult female gonads, respectively, with expanded levels of pMad and loss of Bam expression as well as loss of ECs' extensions in the germarium (*Figures 8H, J and 4F, G*, respectively). In contrast, germline-specific KD or disruption of its activity using other tissue-specific drivers (wing, eye, etc.) resulted in negligible effects (*Figure 8A, C*, and data not shown). This finding confirmed AdamTS-A as a biologically relevant component that likely acts downstream of both Nup107 and Dsx in this context.

We next examined whether overexpression of *AdamTS-A* can mitigate the phenotypes induced by the loss of either *Nup107* or *dsx*$^F$. To that end, we knocked-down either *Nup107* or *dsx* and concomitantly overexpressed *AdamTS-A*, again using *GFP* overexpression as a control. Analysis of the resulting ovarian samples revealed a high proportion of underdeveloped ovaries, similar to flies compromised for either *Nup107* or *dsx*. This suggests that while important, AdamTS-A cannot be the sole player downstream of the Nup107–Dsx axis and there must be other targets which contribute to its function(s).

## Discussion
### Nup107 activity is required in ICs and ECs during ovarian development and oogenesis

We have shown that Nup107 activity in the somatic component of the gonad is necessary for the proper development and function of the ovaries. In which somatic cell type, the activity of Nup107 is necessary? The fact that KD of Nup107 using the *tj-Gal4* driver resulted in larval and adult aberrant phenotypes indistinguishable from those induced by the Nup107 loss of function mutation, indicates that the ovarian function of Nup107 is primarily required in the Tj-expressing cells. Notably, the *tj-Gal4* driver is not expressed in TF cells either at larval or adult stages. TF cells together with the CCs constitute the ovarian stem cell niche. During larval ovarian development, Tj is expressed in the ICs,

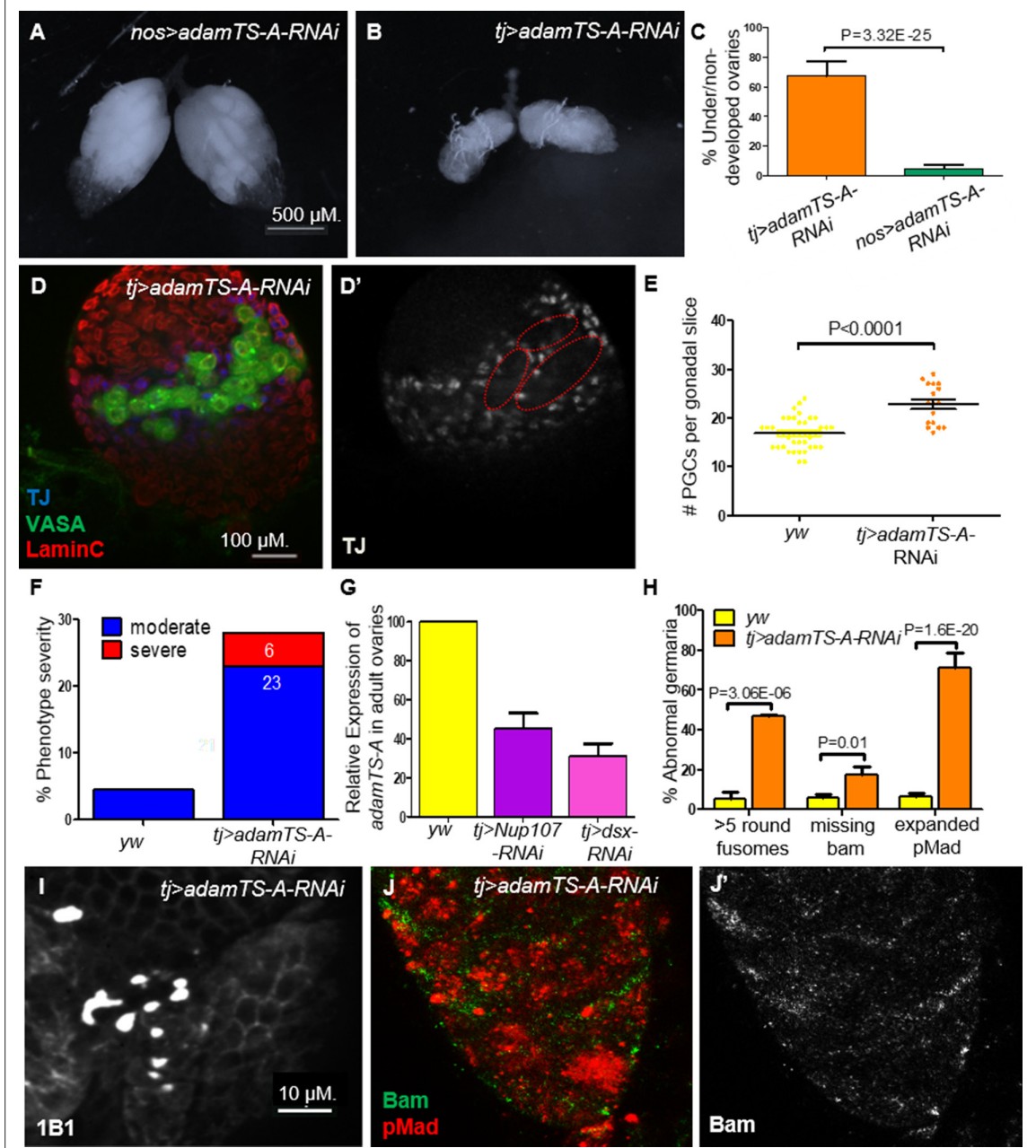

**Figure 8.** *AdamTS-A* KD larval and adult ovaries show aberrant phenotypes. (**A**) Germline KD of *AdamTS-A* (*nos-Gal4*) results in negligible effects, compared to somatic KD (*tj-Gal4*), which results in (**B**) severely underdeveloped ovaries. (**C**) Quantitation of under/nondeveloped ovaries in *tj-* vs. *nos-Gal4*-driven *AdamTS-A* KD (n = 126, 108) flies. (**D**) Somatic KD (*tj-Gal4*) of *AdamTS-A* results in larval gonads containing excess numbers of primordial germ cells (PGCs; VASA, green) and abnormally dispersed intermingled cells (ICs; TJ, blue). (**E**) Quantitation of the total number of PGCs per confocal section in each individual gonad in *yw* (n = 42) and *AdamTS-A* KD (n = 17) larvae. (**F**) The percentage of IC severity phenotypes found in *yw* (n = 22) and *AdamTS-A* KD (n = 17) larval gonads. (**G**) Relative expression of *AdamTS-A* measured by RT-qPCR. (**H**) Quantification of cells with round fusomes (n = 57), Bam expression (n = 145, green), and pMad expression (n = 73, red) in *yw and AdamTS-A* KD ovaries. *tj-Gal4*-driven *AdamTS-A* KD results in (**I**) excess number of cells with spherical fusomes (anti-1B1), (**J**) expanded pMad (red), and (**J'**) reduced Bam expression.

CCs, and FSC progenitors (***Knauff et al., 2009***). CCs, which are derived from the ICs (***D'Angelo, 2018***, ***Wang et al., 2011***), are formed at the base of fully formed TFs at the transition from the third larval instar to prepupal stage (***Ren et al., 2018***, ***Levi et al., 2020***). Therefore, the aberrant PGCs and ICs observed in the larval gonads are not due to impairment of Nup107 or Dsx activities in CCs. Likewise, FSC progenitors are also not the candidate cells for the site of action of Nup107 activity as

they reside posterior to ICs with a minimal physical contact with only a few posteriorly located PGCs (*Slaidina et al., 2020*). Together these data imply that Nup107 acts specifically in ICs enabling them to effectively interact with the PGCs. Consistent with this notion, loss of *Nup107* affected the behavior of ICs such that these cells showed varying degrees of failure to mingle with the PGCs. A severe failure of ICs and PGCs to interact in the larval gonad is expected to cause an ovarian-dysgenesis-like phenotype as it is essential for the germarium development and ovariole formation. Likewise, a milder failure of intermingling in the larval gonad would allow for the formation of adult ovaries. However, in the adult ovary Nup107 activity is further required in ECs for the formation of their cellular extensions and regulation of differentiation of the GSCs. Thus, it will be important to determine in future studies the functional relationship between Nup107 and the signaling pathways which previously were shown to regulate the formation of these cellular extensions.

## How does Nup107 execute its ovarian function?

Our studies have revealed that Nup107, a ubiquitously expressed nuclear envelope protein, is a crucial player during female gonad formation. How does an essential housekeeping protein critical for nuclear transport, perform such a sex- and tissue-specific function(s)? We envisage two possible scenarios, not necessarily mutually exclusive, to explain how the specific mutation in *Nup107* results in ovary-specific aberrant phenotypes. In the first scenario, Nup107 would specifically mediate nucleocytoplasmic translocation of factor(s) or downstream effector(s) required for ovarian development. Indeed, recent studies have demonstrated that Nup107 is involved in translocation of specific factors. For instance, in the event of DNA damage, Nup107 directly interacts with the apoptotic protease activating factor 1 (Apaf-1 also known as *Drosophila* ARK) and mediates its transport into the nucleus to elicit cell-cycle arrest (*Jagot-Lacoussiere et al., 2015*). Furthermore, it has been shown in tissue culture that specific Nucleoporins, including Nup107, are required for nuclear translocation of SMAD1, an important downstream effector of the Dpp/BMP pathway (*Chen and Xu, 2010*).

Alternatively, accumulating evidence has documented that in addition to their primary function in regulating the exchange of molecules between the nucleus and cytoplasm, NPC components may contribute to genome organization and tissue-specific regulation of gene expression (*Raices and D'Angelo, 2017*; *D'Angelo, 2018*) in a nuclear transport-independent manner. Consequently, such moonlighting activities may not be confined to the nuclear envelope which is the primary native location of these proteins. For instance, mammalian Nup107–160 complex (a subcomplex of the NPC of which Nup107 is a key component) has recently been shown to shuttle in and out of GLFG nuclear bodies containing Nup98, a nucleoporin that regulates multiple aspects of gene regulation (*Morchoisne-Bolhy et al., 2015*). Consistently, Nup107 was shown to regulate levels of specific RNAs through gene imprinting (*Sachani et al., 2018*). Furthermore, using an *RNAi*-based assay, Nup107 was identified as a positive regulator of OCT4 and NANOG expression in human ESCs (*Onal et al., 2012*). In this regard, it is noteworthy that the recently published chromatin-binding profile of Nup107 suggested that Nup107 specifically targets active genes (*Gozalo et al., 2020*). Altogether these data support the possibility that Nup107 affects transcription of specific target genes in a tissue- and sex-specific manner either directly or indirectly.

## Master regulator Dsx acts downstream of housekeeping gene Nup107

In *Drosophila melanogaster*, Sex-lethal (Sxl) is the master determinant of somatic sexual identity, regulating a splicing dependent regulatory cascade resulting in the presence of alternatively spliced sex-specific isoforms of Dsx protein, Dsx-F and Dsx-M, in females and males, respectively. Subsequent dimorphic sexual development including sex-specific gonad morphogenesis is under the control of these Dsx isoforms. Consistently, Dsx proteins deploy components of the housekeeping machinery to achieve sex-specific development of the gonads. Thus, such 'maintenance' factors are unlikely to be involved in any regulatory capacity. Our data challenge this notion and demonstrate the presence of sexually dimorphic circuitry downstream of a 'housekeeping' nuclear envelope protein, Nup107, which regulates the expression the female form of *dsx*.

The similar sex-specific and ovary-restricted phenotype associated with compromised Nup107 activity in both humans and flies implies common underlying molecular mechanisms. We have identified Dsx as the primary target acting downstream of Nup107 in *Drosophila* ovarian development. The mammalian homologues of Dsx, the Dmrt family of transcription factors, also function during

sex-specific gonad development. However, in mammals the main function of *Dmrt* genes in the gonad is to promote male-specific differentiation. While detailed functional analysis is not available, it is plausible that in mammals, another key female-specific transcription factor, like *Foxl2* (*female-specific forkhead box L2*) may act downstream of Nup107 to substitute for DsxF in flies.

## Dsx is essential for proper ovarian development

An interesting study by Van Doren et al. showed that the key components of the stem cell niche, that is the hub in males and the TFs in the case of females, are still formed in the absence of *dsx*, but this happens in a stochastic manner in both XX and XY *dsx* null mutant individuals (*Camara et al., 2019*). These results indicate that in the context of the developing stem cell niche, Dsx may not act in an instructive manner, but is instead required to ensure that the proper program (male or female) is selected, which does not require Dsx activity for the execution of subsequent sex-specific development. Nevertheless, their findings clearly demonstrated that the resulting adult ovaries and testes are improperly formed, consist of aberrant structures, arguing that Dsx activity, is critical for proper gonad development outside of the stem cell niche. Our observations are consistent with this suggestion. Moreover, our experimental strategies and results differ in two important ways. First, our experiments have relied on reduction of only the female form of *dsx* that is *dsx-F* which allowed for sex determination and thus we did not observe 'male' structures or cellular identity transformations. Second, in our experiments we use *tj-Gal4* driver which is not expressed in the stem cell niche (TF cells) but in other somatic gonadal cells. This experimental design enabled us to uncover a novel developmental function of Dsx-F in ICs and their adult descendants, ECs. Supporting this notion Van Doren and Oliver labs showed that *RNAi* KD of *dsx* also resulted in small ovaries (*Clough et al., 2014*).

## AdamTS-A modulates BMP signaling via regulation of EC extensions

We have found that the secreted metalloprotease AdamTS-A is an important downstream component in the Nup107–Dsx axis, as KD of *AdamTS-A* results in phenotypes similar to those elicited by loss of either *Nup107* or *dsx*. In the adult ovary, these aberrant phenotypes include loss of EC membrane protrusions and expanded BMP signaling. This raised the question of how AdamTS-A regulates the range of BMP signaling. The ECM, which is produced and secreted by cells, has the structure of a complex fibrillar meshwork and provides structural support and tissue integrity, playing an active role in regulating cell behavior (*Rozario and DeSimone, 2010*; *Brown, 2011*; *Wolf and Friedl, 2011*). ECM proteoglycans sequester and modulate chemical signals, including growth factors and guidance molecules. Furthermore, type IV collagens, major components of the ECM, were shown to restrict Dpp signaling in the ovary (*Wang et al., 2008*). This is particularly intriguing, since in *C. elegans* Gon-1, the homolog of AdamTS-A, was shown to genetically interact with a type IV collagen (EMB-9) in the regulation of gonadogenesis (*Kubota et al., 2012*). This raised the possibility that AdamTS-A, secreted by ECs, restricts Dpp movement in the germarium through cleavage of ECM components. However, by knocking down *coracle* in ECs, we have shown that disruption of their cellular protrusions, which encapsulate the germ cells, leads to expansion of BMP signaling. This implies that the activity provided by these cellular extensions is necessary and sufficient for restricting the BMP signal.

Thus, Nup107, Dsx, and AdamTS-A all function in ECs and are necessary for the formation and maintenance of the cellular protrusions which are required for restricting the BMP signal emanating from the GSC's niche. Further, it appears that Adam-TS-A activity in the ECM is required for the formation and/or maintenance of these cellular protrusions. Our results indicate that in this context AdamTS-A regulates BMP signal distribution indirectly via regulation of the cellular protrusion maintenance. It is also possible that AdamTS-A utilizes these cellular extensions in order to reach the ECM away from the ECs in the GSC region, where it acts to restrict Dpp trafficking.

## Conclusions

Overall, our results support a model where Nup107 regulates the expression of *dsx*, either directly or indirectly, while Dsx directly regulates the transcription of multiple target genes including *AdamTS-A*. Our observations have also uncovered that Dsx$^F$ controls somatic niche function by calibrating the range and/or strength of Dpp/BMP signaling, possibly via modulation of the level and/or activity of the ECM components. Thus, it will be critical to elucidate how activities of nonsex-specific components such as Nup107 are coordinated with sex-specific regulation to achieve the precise specification

and patterning underlying gonad development. This is of particular significance since modulation of BMP signaling circuitry is inextricably linked with the establishment and maintenance of stem cell fate. Importantly, as in the case of Nup107, BMP signaling is also required in a nonsex-specific manner in a variety of developmental contexts.These observations therefore open new avenues toward the critical examination of how a productive molecular dialog is established between nonsex-specific house-keeping machinery and versatile intersecting developmental pathways, in order to ultimately achieve proper sex-specific gonadogenesis crucial for fertility, and transmission of genetic information.

## Materials and methods

### Fly strains

Flies were raised and maintained at 25°C on cornmeal yeast extract media (6 g methyl paraben, 24 g bactoagar, 3200 ml water, 78 g Brewer's yeast, 224 g cornflour, 75 g sugar, 165 ml molasses, 48 ml propionic acid). *yw* was used as a wild-type strain. The generation of *RFP-Nup107*$^{WT}$ and *RFP-Nup107*$^{D364N}$ transgenic flies was previously described in **Weinberg-Shukron et al., 2015**. *nanos-Gal4* and *tj-Gal4* were gifted by Lilach Gilboa's lab (Weizmann Institute of Science, IL). *Nup107 RNAi* and *AdamTS-A RNAi* lines were provided by the Vienna *Drosophila* Research Center (VDRC, Vienna, Austria #108,047 and #110,157). *dsx RNAi* and *dsx UAS, Cora RNAi* provided by Bloomington *Drosophila* Stock Center (BDSC; Indiana University; USA; #35,645, #41,864 #44,223, #51,845, and #3500). *tub-Gal80*$^{ts}$ was gifted by Estee Kurant's lab (Department of Human Biology Faculty of Natural Sciences University of Haifa, IL), *c587-Gal4; UAS-GFP* was gifted by Hila Toledano's lab (Department of Human Biology, University of Haifa, IL), *UAS -AdamTS-A* was gifted by Deborah J Andrew's lab (Department of Cell Biology, The Johns Hopkins University School of Medicine, Baltimore, USA).

The lines' complete genotypes:

1. *yw: y¹w\*;;*
2. *RFP-Nup107*$^{WT}$*: y¹w\*; Nup107*$^{E8}$*/Nup107*$^{E8}$*; RFP-Nup107*$^{WT}$
3. *RFP-Nup107*$^{D364N}$*: y¹w\*; Nup107*$^{E8}$*/Nup107*$^{E8}$*; RFP-Nup107*$^{D364N}$
4. *nanos-Gal4: w\*;; nos-GAL4::VP16*
5. *tj-Gal4: w\*; tj-Gal4/Cyo;*
6. *Nup107 RNAi: y¹ w\*; UAS-Nup107-RNAi;*
7. *AdamTS-A RNAi: w\*; UAS-AdamTS-A RNAi;*
8. *dsx RNAi: y¹; UAS-dsx-RNAi/Cyo; or y¹;; UAS-dsx-RNAi*
9. *UAS dsx: y¹ w\*;; UAS-dsx;*
10. *UAS AdamTS-A: y¹ w\*;; UAS-AdamTS-A;*
11. *c587-Gal4; UAS-GFP*
12. *tub-Gal80*$^{ts}$*: Sco/Cyo; tub-Gal80*$^{ts}$*/ =*

### The crosses genotype

1. *tj-Gal4 > Nup107 RNAi:; tj-Gal4/UAS-Nup107-RNAi;*
2. *nanos-Gal4 > Nup107 RNAi:; UAS-Nup107-RNAi; nos-GAL4::VP16*
3. *tj-Gal4 > AdamTS A RNAi:; tj-Gal4/UAS-AdamTS-A RNAi;*
4. *tj-Gal4 > dsx RNAi:; tj-Gal4/UAS dsx-RNAi; or w; tj-Gal4; UAS-dsx-RNAi*
5. *tj-Gal4 > Nup107 RNAi, UAS dsx:; tj-Gal4/UAS-Nup107-RNAi; UAS dsx*
6. *tj-Gal4 > Nup107 RNAi, UAS AdamTS-A:; tj-Gal4/UAS-Nup107-RNAi; UAS AdamTS-A*
7. *c587-GFP: c587-Gal4; UAS-GFP/+;*
8. *c587-Gal4 > Nup107 RNAi: c587-Gal4; UAS-GFP/UAS-Nup107-RNAi;*
9. *c587-Gal4 > Cora RNAi1: c587-Gal4; UAS-GFP/UAS-Cora-RNAi;*
10. *c587-Gal4 > Cora RNAi2: c587-Gal4; UAS-GFP/+; +/Cora-RNAi*
11. *tj*$^{ts}$*: tj-Gal4/Cyo; tub-Gal80*$^{ts}$

### Adult and larval gonad dissections

Stage LL3 larvae were collected and subsequently dissected for their gonads according to **Maimon and Gilboa, 2011**. Adult ovaries were dissected from 3- to 5-day-old females placed on yeast for 24–36 hr in the company of males. All experiments were performed at 25°C, and all were independently

repeated at least twice. Dissection was performed in Ringer's solution (130 mM NaCl, 5 mM KCl, 2 mM CaCl$_2$, 50 mM Na$_2$HPO$_4$, pH 7).

## Immunostaining and imaging

Fixation and immunostaining of larval gonads or adult ovaries were carried as previously described by *Maimon and Gilboa, 2011* or by *Preall et al., 2012*. In brief, gonads or ovaries were fixed in freshly prepared 5% paraformaldehyde (Electron Microscopy Sciences; Cat# 15714) for 30 min at room temperature. Blocking was carried out in wash buffer supplemented with 1% bovine serum albumin (BSA; MP Biomedicals; Cat. #160069). Primary antibodies were diluted and incubated overnight at 4°C in wash buffer supplemented with 0.3% BSA. The following primary antibodies were used: guinea-pig anti-Tj (1:10,000; gifted by Dorothea Godt's lab at the University of Toronto, Toronto, Canada), rat IgM anti-VASA (1:100; Developmental Studies Hybridoma Bank [DSHB, Iowa City, IA, USA]), mouse anti-Hts (1B1; 1:20; DSHB), mouse anti-Bam (1:50; DSHB), mouse anti-Cora (C566.9; 1:100; DSHB), rabbit anti-smad3 (1:100; abcam, Cambridge, MA, USA; #ab52903), rabbit anti-cleaved caspase3 (1:200; Cell Signaling Technology, CST, Danvers, MA, USA; #9661), and mouse anti Lamin (1:20; gifted by Yosef Gruenbaum's lab at the Hebrew University of Jerusalem, IL). Secondary antibodies (1:400) were conjugated to either Cy2, Cy3, or Cy5 (Jackson Immuno Research Laboratories; West Grove, PA, USA). Ovaries were mounted in Vectashield (Vector Laboratories; Burlingame, CA, USAVE-H-1000). Images were taken on a TE2000-E confocal microscope (Nikon) using ×20 or ×60 objectives, occasionally with an additional ×1.5 zoom. Figures were edited using Adobe Photoshop CC 2017.

## qRT-PCR analysis

Total RNA was isolated from larval gonads at stage LL3 (add symbol 80 per sample) of *yw*, *Nup107$^{WT}$*, and *Nup107$^{D364N}$* using the RNeasy mini kit (Qiagen Valencia, CA, USA; #74104). Briefly, wandering third instar larvae were collected, females were selected and dissected into tubes in liquid nitrogen and lysis buffer was added. Total RNA was extracted as per the manufacturer's instructions. cDNA was made using the high-capacity cDNA reverse transcription kit (Applied Biosystems, Foster City, CA, USA) using an equal amount of total RNA from each sample. Real-time q-PCR analyses were carried out using the Powersyber Green PCR Master Mix and QuantStudio 12 k flex (Applied Biosystems). *Rsp17* and *TBP2* served as reference genes using the comparative Ct method. Each sample was analyzed in triplicate; results were confirmed by at least two independent experiments. Primer sequences (from HyLabs, Israel, LTD) used for qPCR were: dm_*Nup107*_F, 5'- *GCCA AGCAAACCATCGAACTC*-3'; dm_ *Nup107*_R, 5'- *GCAGTAGGCGATGATCCCAG*-3'; dm_*doublesex*_F, 5'-*TTGCCGATCTCAGTTTCCGT*-3'; dm_*doublesex*_R, 5'- *GCTCCCAAGGATAGCGGAAT*-3'; dm_*AdamTS-A*_F, 5'- *GGGAATGAGCCGAACAAGAC*-3'; dm_*AdamTS-A*_R, 5'- *AAGTTCTGGTCG GGATAGCC*-3'.

## Statistical analysis

The number of under/nondeveloped adult ovaries in wild-type and mutant flies, the varied expression of 1B1, Bam, Cora, as well as the number of spherical fusomes in germaria were compared pairwise using Fisher's Exact Test for 2 × 2 tables. The raw two-tail p values were adjusted for the multiple comparisons using either the Bonferroni correction or Holm's modification (*Holm, 1979*) thereof, as appropriate. p values of less than 0.05 were considered significant.

The number of PGCs, ICs, and fusomes in groups of larval gonads was compared using a Kruskal–Wallis test (*Kruskal-Wallis, 2013*). In experiments where the differences among the groups were found to be significant (K–W p value <0.05), pairwise comparisons were carried out using Conover's post hoc test (*Conover, 1999*).

The statistical significance of the observed overlap between our Nup107 target genes list and the previously reported Dsx targets was calculated using the hypergeometric test (https://systems.crump.ucla.edu/hypergeometric/index.php). The specific parameters were as follows: number of successes $k = 47$; sample size $s = 82$; number of successes in the population $M = 3717$; population size $N = 15,835$. The results obtained were: expected number of successes = 19.2481212503947. The results are enriched 2.44-fold compared to expectations hypergeometric p value = 4.4742336544062e−11.

## RNA-seq

Total RNA was isolated from *yw*, *RFP-Nup107*[WT], and *RFP-Nup107*[D364N] *Drosophila* larval gonads following dissection at stage LL3 using the RNeasy Kit (Qiagen) according to the manufacturer's protocol. RNA purity and concentration were determined by T-042 NanoDrop Spectrophotometer (Thermo Fisher Scientific Inc, Waltham, MA, USA) and integrity by 2100 Bio-analyzer (Agilent Technologies, CA, USA). Total RNA was reverse transcribed to cDNA using SENSE Total RNA-Seq Library Prep Kit for Illumina (Lexogen, Vienna, Austria), according to the manufacturer's protocol, with poly-A selection. Libraries were multiplexed and sequenced on Illumina's NextSeq 500 machine, with a configuration of 75 cycles, single read. Raw reads were processed to remove low quality, error prone, and adapter sequences, according to Lexogen's SENSE libraries recommendations. High-quality reads were aligned to the fly genome, assembly BDGP6, that was supplemented with the sequences of *GFP* and the *RFP-Nup107* constructs. Alignment was performed with TopHat, allowing for up to 5 mismatches per read. Differential expression analysis, for all genes from release 84 of the Ensembl database, was performed with the DESeq2 package, using default parameters, including the threshold for significance that was $p_{adj}$ <0.1. Significant genes were further filtered to include only genes whose up- or downregulation was greater in the mut/yw comparison than in the wt/yw one. For that end, a difference of at least 0.1 between the absolute log2FoldChange (mut/yw) and absolute log2FoldChange (wt/yw) was used as the filtering threshold. All raw data, as well as software versions and parameters, have been deposited in NCBI's Gene Expression Omnibus (*Edgar et al., 2002*) and are accessible through GEO Series accession number GSE141094.

## Acknowledgements

We thank L Gilboa, E Kurant, H Toledano, D J Andrew, D Godt, Y Gruenbaum, the Developmental Studies Hybridoma Bank, the Vienna *Drosophila* RNAi Center, and the Bloomington *Drosophila* Stock Center for reagents and fly stocks. We thank Z Paroush for critical reading and comments on the manuscript. We thank N Grover for assistance with statistical analysis. We also thank M Korner, M Bronstein, and T Schnitzer-Perlman at the Center for Genomic Technologies at the Hebrew University of Jerusalem for their assistance and expertise in RNA-seq. Funding: This study was supported by the Legacy Heritage Biomedical Program of the Israel Science Foundation (grant 1788/15 to O Gerlitz and D Zangen), by the Israel Science Foundation (grant 1814/19 to O Gerlitz, grant 2295/19 D Zangen). G Deshpande was supported by the National Institute of Health grant awarded to P Schedl and G Deshpande (NICHD:093913). Tgst Levi's work was also supported by stipends from the Ministry of Science & Technology, Israel, and by the Ministry of Aliyah & Integration, Israel.

## Additional information

### Funding

| Funder | Grant reference number | Author |
| --- | --- | --- |
| Israel Science Foundation | 1788/15 | David Zangen Offer Gerlitz |
| Israel Science Foundation | 2295/19 | David Zangen Offer Gerlitz |
| National Institutes of Health | 093913 | Girish Deshpande |
| Ministry of Science and Technology | | Tgst Levi |
| Ministry of Aliyah and Integration | | Tgst Levi |

The funders had no role in study design, data collection, and interpretation, or the decision to submit the work for publication.

## Author contributions
Tikva Shore, Data curation, Formal analysis, Funding acquisition, Investigation, Methodology, Validation, Visualization, Writing - original draft, Writing - review and editing; Tgst Levi, Data curation, Formal analysis, Funding acquisition, Investigation, Methodology, Validation, Visualization, Writing - review and editing; Rachel Kalifa, Amatzia Dreifuss, Dina Rekler, Ariella Weinberg-Shukron, Tzofia Bialistoky, Victoria Moyal, Shira Leebhoff, Formal analysis, Investigation; Yuval Nevo, RNA-Seq bioinformatic analysis, Validation; Merav Yaffa Gold, Investigation, Writing - review and editing; David Zangen, Funding acquisition, Resources; Girish Deshpande, Conceptualization, Funding acquisition, Visualization, Writing - original draft, Writing - review and editing; Offer Gerlitz, Conceptualization, Data curation, Formal analysis, Funding acquisition, Investigation, Methodology, Project administration, Supervision, Validation, Visualization, Writing - original draft, Writing - review and editing

## Author ORCIDs
Tikva Shore ⓘ http://orcid.org/0000-0002-7140-0226
Tgst Levi ⓘ http://orcid.org/0000-0001-9221-1873
Merav Yaffa Gold ⓘ http://orcid.org/0000-0002-9978-2262
Girish Deshpande ⓘ http://orcid.org/0000-0002-5200-7090
Offer Gerlitz ⓘ http://orcid.org/0000-0002-1574-2088

## Decision letter and Author response
Decision letter https://doi.org/10.7554/eLife.72632.sa1
Author response https://doi.org/10.7554/eLife.72632.sa2

# Additional files

## Supplementary files
• Transparent reporting form

• Supplementary file 1. Processed data results of transcriptomic analysis performed on $Nup107^{WT}$ and $Nup107^{D364N}$ larval gonads.

• Supplementary file 2. List of 82 candidate genes with differential expression following Nup107 loss identified in transcriptomic analysis, with emphasis on those found to be Dsx targets.

## Data availability
All raw RNA-seq data, as well as software versions and parameters, have been deposited in NCBI's Gene Expression Omnibus and are accessible through GEO Series accession number GSE141094.

The following dataset was generated:

| Author(s) | Year | Dataset title | Dataset URL | Database and Identifier |
|---|---|---|---|---|
| Shore T | 2019 | Nucleoproin 107 Mediates Femae Sexual Differentiation via Doublesex | https://www.ncbi.nlm.nih.gov/geo/query/acc.cgi?acc=GSE141094 | NCBI Gene Expression Omnibus, GSE141094 |

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
