## [Editor Report]

This study uses *Drosophila* as a model to study a specific mutant in a gene encoding a nuclear pore protein, whose counterpart in human leads to a rare disease called XX-ovarian-dysgenesis. Intriguingly, the fly mutants mimic the syndromes identified in human patients, such as failures in ovary development and function. The authors use fly ovary as a model to study the reasons underlying the phenotypes, which should provide insight to our understanding of this disease.

---

## [Decision Letter]

**Decision letter after peer review:**

Thank you for submitting your article "Nucleoporin107 mediates female sexual differentiation via *Dsx*" for consideration by *eLife*. Your article has been reviewed by 3 peer reviewers, and the evaluation has been overseen by a Reviewing Editor and Utpal Banerjee as the Senior Editor. The following individual involved in review of your submission has agreed to reveal their identity: Yukiko M Yamashita (Reviewer #3).

This manuscript uses genetics, cell biology and genomics tools to study the loss-of-function phenotypes of the Nucleoporin107 (Nup107) gene. This work showed that Nup107 acts in the somatic gonadal cells to regulate BMP signaling and proper germline differentiation in *Drosophila* ovaries. Interestingly, these phenotypes resemble the syndromes in human XX-ovarian-dysgenesis patients. Using RNA-seq, the authors identified targets genes whose transcript levels changes in the nup107 mutants. Among them, two are considered as key targets, including *dsx* and AdamTS-A. Genetics analyses demonstrate similar phenotypes when compromising *dsx* or AdamTS-A in somatic gonadal cells; and for *dsx*, overexpression of it can rescue Nup107 knockdown phenotypes. Overall, these results should be of interest to researchers in germ cell biology and developmental biology fields. The presented results are well done and the data analyses are proper. However, at several places in order to reach the conclusions, more rigorous experiments should be performed (see revision details). This study, at the current format, is short of mechanistic insight how Nup107 function in the ovaries. In addition, the reviewers noticed inconsistent results in this work compared to previous published studies, such as the roles of *Dsx* and the consequences caused by high BMP signaling in the ovaries. In conclusion, this work has the potential but requires essential additional data to support the central claims.

Essential revisions:

Shore et al. studied nup107 mutant and somatic knockdown phenotypes in the *Drosophila* ovaries. Most of the data of this manuscript are presented in a clear and logic manner. The major concerns include:

1. The major concern is the lack of mechanistic insight of Nup107. For example, how does Nup107 regulates *dsx* and AdamTS-A transcription? Is this role related to being part of the nuclear pore complex or in an unconventional manner?

2. Another major concern is the inconsistency with published studies for *Dsx*, where *dsx* mutant or knockdown show sex determination defects with intersexual phenotypes vs. in this paper no such phenotype was detected (Figure 5).

3. Some of the presented results are confusing: for example, data shown in Figure 6F indicate quite normal egg chambers therefore little differentiation defects, which are in contrary with their previous conclusion that Nup107 is required for proper female GSC differentiation. All these places need further clarifications.

4. Most of the phenotypes use RNAi, however, more controls are needed to show the efficiency of the RNAi. For example, in the germline, the authors should show nanos-Gal4 driven RNAi does knockdown target gene expression. In the somatic cells, the authors should add the temporal control of the knockdown experiments to understand the developmental stage when Nup107 is needed.

5. Similarly, for the rescuing experiments, which are critical for drawing quite a few critical conclusions for this paper, proper control to avoid interpretation caveats should be included.

6. The reviewers also noticed quite some critical literatures regarding female gonadal development are missing.

*Reviewer #1 (Recommendations for the authors):*

Overall the studies were well done and the analyses are rigorous, here are some suggestions that will hopefully improve this study and make some conclusions more solid:

General suggestions:

1. For the cell type specificity for Nup107, I think in addition to the cell type-specific RNAi experiments, it would be more complete to add the cell type-specific rescuing experiments and results. For example, expression of Nup107 in Tj-positive somatic gonadal cells in the nup107 mutant background.

2. In several figures, nos-Gal4 driving RNAi does not yield to significant phenotype but tj-Gal4 does, I think it is important to show that the germline RNAi does lead to reduced levels of the target genes in germ cells.

3. It has not been mentioned about the basic cellular mechanism of the functions of Nup107. For example, which cell type is Nup107 expressed and what is its subcellular localization? Does these patterns change in the mutant background? In the Discussion, it was mentioned that Nup107 is a "ubiquitously expressed nuclear envelope protein", but please show the results.

Specific suggestions on results and data analyses:

4. Figure 1, Nup107 RNAi, need to quantify the KD efficiency. Also, in Figure 1G, what do the average and deviation stand for as the Y-axis is for the percentage, is it from a number of independent experiments? Also, in the figure legend, need to clarify what statistical method was used for the P values. Scale bars are also needed for panel B, C, D, E, and F.

5. Regarding this statement "Thus, GSCs can be readily identified by the presence of a round fusome, whereas their differentiated daughter cells display extended, branched fusomes." In fact, both GSCs and CBs have a round spectrosome, while more differentiated cystocytes have branched fusomes.

6. Given both GSCs and CBs have spectrosomes and each ovariole contains 2-3 GSCs and probably around 2-3 CBs, I wonder whether for this statement: "Ovary staining revealed that unlike yw and Nup107WT germaria, which contained 2-3 spherical fusomes", the number would be a big larger than 2-3, and if this is true, then "an average of 6 spherical fusomes per germarium" may not be a phenotype?

7. Figure 3G quantifies germaria with few/no Cora extensions, as few is kind of subjective, it would be clearer to clarify the threshold for calling "few" in the figure legend. The same comments apply for Figure 5D quantification.

8. Please add more statistical analyses for panel 4D and 4J.

9. For RNA-seq, please show more analyses including biological replicates of the three different genotypes. Also, the authors mentioned that they used "LL3 larval gonads" for RNA-seq, I assume they mean larval ovaries but please specify it in the Results.

10. Regarding the experiments to show "*Dsx* overexpression rescues the phenotypes induced by Nup107 loss", I think it is better to rescue driving *Dsx*_F_ using tj-Gal4 at the nup107 mutant background. The co-expression of both *Dsx*_F_ and nup107 RNAi using the same tj-Gal4 driver could result in a "rescue" due to competition for the same driver. Therefore, either rescue using the nup107 mutant background or for the rescue using tj> *Dsx*_F_ + nup107 RNAi, provide the control with tj> lacZ (or another non-specific UAS-transgene) + nup107 RNAi.

11. Regarding "AdamTS-A acts downstream of Nup107 and *Dsx*", does overexpression of AdamTS-A also rescue nup107 mutant phenotypes?

*Reviewer #2 (Recommendations for the authors):*

1. The authors perform somatic cell manipulation using tj-Gal4 for a large majority of their experiments and observe defects in 3-to-5-day old females. Therefore, defects that are observed in the ovary could be due to transgene expression during adulthood (which the authors acknowledge in lines 388-392) of the manuscript. A simple test would be to combine tj-Gal4 with tubPGal80ts (a temperature-sensitive allele of Gal80) and perform the Nup107 knockdown and determine whether there are ectopic GSCs similar as without the temporal control.

2. It was previously shown that loss of Tkv in the developing soma results in ectopic GSCs in the adult ovary (Tseng et al., 2018), similar to results observed with loss of Nup107, *dsx*, and AdamTS-A (this study). Is BMP signaling in the developing gonad also disrupted?

3. The authors propose that Nup107 is required for proper differentiation of escort cells. How is that determined when there were similar numbers of escort cells in tj-Gal4 > Nup107 RNAi compared to control. Loss of escort cell protrusions may not directly impact differentiation as there are many additional factors that can influence escort cell protrusions such as defects in EGFR signaling and changes in diet.

4. The authors show images for cleaved caspase (Figure 6F-G), but there is not any explanation in the text. Are the authors proposing an additional requirement for Nup107 in older egg chambers?

5. The conclusion that AdamTS-A is a direct target of Nup107 and *dsx* would be greater strengthened by rescue analysis (overexpression of AdamTS-A in a tj-Gal4 > Nup107 or *dsx* RNAi).

6. The authors should indicate how region 2a in the germarium is being determined.

7. The 1B1 staining in Figure 2E is hard to see. Is there a better representative image that could be used?

8. It should be indicated throughout the text that BMP signaling and other defects observed in the adult germaria and ovary could be due to either developmental defects or adult-specific defects.

9. What are "high" spherical fusomes (line 276)?

10. There isn't a "standard cornmeal yeast extract media" (see Lesperance and Broderick, 2020). Please provide the ingredients and recipe used for food. Also, please indicate whether standard food was supplemented with wet yeast paste.

*Reviewer #3 (Recommendations for the authors):*

– Nup107 and *Dsx* mutants show degenerated ovary, and they show that this is likely due to differentiation block (with high pMad in many germ cells). However, as can be seen in Figure 2, the ovarioles seem to have normal-looking assembly lines of egg chamber, and I wonder what is the exact cause of ovary degeneration. Can you please add a bit more details of exactly how stem cell differentiation defects lead to ovary degeneration in Nup107 or *Dsx* depletion?

– One cannot call *Dsx* as a 'target' of Nup107. What is the mechanism by which *Dsx* expression is changed in nup107 mutant?

– Is RNAi efficiency validated for all RNAi lines?

– What I am most confused about is the apparent inconsistency of the *Dsx* mutant phenotype described in this study vs. previous studies (Erika Bach, Mark van Doren and Erika Matunis' lab). Perhaps the data are not inconsistent, but previous work focused on sex transformation, and I am puzzled why the ovary of *Dsx* mutant described here looks quite normal in terms of sex identity. And it would be extremely helpful if the authors can place their data in the context of the previous work.

---

## [Author Response]

Essential revisions:Shore et al. studied nup107 mutant and somatic knockdown phenotypes in the *Drosophila* ovaries. Most of the data of this manuscript are presented in a clear and logic manner. The major concerns include:1. The major concern is the lack of mechanistic insight of Nup107. For example, how does Nup107 regulates dsx and AdamTS-A transcription? Is this role related to being part of the nuclear pore complex or in an unconventional manner?

This is an important point and we thank the editors and reviewers for highlighting this issue concerning mechanism. It is important to remember that our studies have identified *Dsx*-F as an important and biologically relevant target/mediator of a nuclear envelope protein Nup107 which is part of the housekeeping machinery primarily required for nuclear transport. Recent studies have also ascribed a non-canonical role in transcriptional regulation to some of the members of this family of proteins including Nup-107. And thus, it will be of great interest to elucidate the precise function of Nup107 in regulating transcription and understand the possible direct as well as indirect ways it may employ to influence transcription of different genes including *Dsx-F*. As the reviewers correctly pointed out, we have not yet uncovered the specific molecular/biochemical activity that enables Nup107 to execute this function. However, we have identified *Dsx*-F as a critical, and sex-specific mediator of Nup107 function. We believe that this is an important and novel point of general significance and future experiments should focus on elucidation of precise mechanism underlying this regulatory activity of Nup107. These efforts will hopefully also shed light on ‘direct’ vs. ‘indirect’ nature of its function, Accordingly, we have elaborated on some of these possibilities in the discussion.

In sum, while our data do not describe the mechanistic underpinnings of this transcriptional regulation, we have uncovered the genetic network which explains how Nup107 executes its ovarian function/activity via *Dsx* and other related ‘targets’ discovered in our analysis including *adamTS-A*.

2. Another major concern is the inconsistency with published studies for Dsx, where dsx mutant or knockdown show sex determination defects with intersexual phenotypes vs. in this paper no such phenotype was detected (Figure 5).

This is another important concern expressed by both the editor and the reviewers. In our view, these differences in the outcomes are most likely a result of design of the individual experiments. In this regard, it is critical to consider the pleiotropic and complex requirement of ‘master’ switch genes such as *Dsx*. Consequently, compromising the activity of such regulators in a tissue-cell type specific manner can result in superficially inconsistent, and at times, even contradictory consequences. In the following we will consider a few scenarios previously reported in the literature that appear to be disparate (and were referred to by the reviewers). It should be noted that this description is somewhat anecdotal and not exhaustive but we hope that the explanation should suffice to alleviate the important concern regarding the discrepancies that confounded the reviewers and editor alike.

Camara et al. demonstrated that *Dsx* is required at the cellular level i.e. in a cell autonomous manner to prevent random switching of the sexual identity of the individual stem cell niches. Consistently, loss of *dsx-F* resulted in inappropriate specification/conversion of “male” cells in female ovaries. Reciprocally, compromising *dsx-M* led to switching of sexual identity in the male gonads (Camara et al., 2019).

However, their findings demonstrated that *Dsx* is critical for proper gonad development since *dsx* mutant adult ovaries (of either XX or XY flies) are clearly defective and underdeveloped (please see figure 1 C, E, H and I in their paper).

Altogether, their findings clearly demonstrated that the resulting gonads are improperly formed i.e. adult ovaries and testes consist of cells of mixed sexual identity arguing that both *dsx-F* and *dsx-M* activities are critical for proper gonad development in both sexes, a finding that is broadly consistent with our data.

Our results, however, differ in two aspects. First, Camara et al. used the dsx1 allele, an amorphic/null allele, which completely removes the function of *Dsx*. In contrast, compromised Nup107 or KD of *dsx* only reduced the levels of *Dsx*, thus allowing for sex determination and this is likely the reason that we observe no “male” structures or cellular identity transformation.

Second, in our experiments we have employed *tj-Gal4* driver which is not expressed in the stem cell niche (TF cells) but is able to drive transcription in other subtypes within somatic gonads. This specific manipulation allowed us to uncover a cell type specific developmental function of *Dsx* in the intermingled cells and their adult descendants*.* Supporting this notion studies from the Van Doren and Oliver labs showed that RNAi KD of *dsx* using *dsx*-Gal4 results in small ovaries. These data are cited in our manuscript (Clough E. et al., Dev Cell. 2014).

By contrast, the results regarding chinmo obtained by the Matunis lab, (Ma et al., Dev Cell. 2014) are not especially relevant to our studies (see below). This is in part due to the fact that loss of *chinmo* leads to generation of female splice isoform of *tra* (tra-F) which in turn leads to production of *dsx-F* and elimination of *dsx*-M, resulting in sex reversal, as shown by Bac lab (Grmai L. et al., PLoS Genet 2018).

Ma et al. engineered RNAi KD of *dsx* only in adult males in the Cyst stem cells (CySC) of the testes (Figure S3). They found that after knocking down *dsx*-F expression for two weeks, the morphologically normal looking testes contained over-proliferating early germ cells which were arrested as early spermatogonia. In addition, these testes contained abnormal aggregates of FasIII positive cells at the periphery that resemble follicle cells in this regard. So unlike loss of *chinmo*, loss of *dsx-F* in CySC did not lead to sex transformation of adult stem cells, i.e, feminization of CySC cells and their descendants. It is also noteworthy that the changes they observed were in the ‘male to female’ direction. Our studies differ in fundamental ways compared to these observations. First, we have analyzed and reported consequences of *dsx* knock down only in females. Also we did not observe any male-like sex transformations or structural alterations and thus we did not probe this issue further by specifically examining for male-specific molecular or protein markers. Rather, our study explores the roles of Nup107 and *Dsx*-F in female ovarian development and deployment of BMP signaling, thus in our opinion the issue of maintenance of sexual identity in adult testes is not pertinent.

3. Some of the presented results are confusing: for example, data shown in Figure 6F indicate quite normal egg chambers therefore little differentiation defects, which are in contrary with their previous conclusion that Nup107 is required for proper female GSC differentiation. All these places need further clarifications.

The effect of loss of *Nup107* activity on the ovarian morphology and function is complex and it leads to highly variable phenotypic consequences. We still don’t fully understand why this is the case and more in-depth analysis will be necessary to explain the variable penetrance and/or expressivity. Supporting this conclusion, we have presented a detailed quantitative analysis of different phenotypes induced by the loss of Nup107 function (Figure 1). These include a variety of ovarian abnormalities ranging from severe dysgenesis (up to 50%) to normal sized ovaries (mild infertility) likely due to defective oogenesis.

However, it should be noted that the relatively normal looking ovaries still consist of several defects which is revealed upon closer examination. Consistently, in our previous publication we showed that the ovarioles from such infertile females with relatively normal sized ovaries are in fact full of aberrant egg chambers. As a result, such ovaries exhibit extensive disintegration of the individual egg chambers, unlike the wild type controls. The other phenotypes seen include condensation of DNA in the nurse cells which is accompanied by apoptotic cell death (Weinberg-Shukron et al., “A mutation in the nucleoporin-107 gene causes XX gonadal dysgenesis.” J Clin Invest. 2015).

In figure 7F we show an example of excess apoptosis in Nup107 KD ovarioles and in figure 7G we show how *Dsx* overexpression rescues this phenotype.

4. Most of the phenotypes use RNAi, however, more controls are needed to show the efficiency of the RNAi. For example, in the germline, the authors should show nanos-Gal4 driven RNAi does knockdown target gene expression. In the somatic cells, the authors should add the temporal control of the knockdown experiments to understand the developmental stage when Nup107 is needed.

This was a useful suggestion and all the reviewers wanted us to confirm this. We have now demonstrated the efficiency of target genes KD using qRT-PCR (see new Figure 1—figure supplement 1).

In addition, in order to identify the developmental stage where Nup107 activity is required, we performed temporal knockdown of Nup107 using a combination of *tj-Gal4* and a temperature sensitive form of the Gal4 repressor, Gal80ts: We allowed for normal ovarian development until hatching, at a non-permissive temperature, and only then knocked down Nup107 in ECs. Subsequent analysis of 4-5 days old female flies was performed. See new addition to manuscript, “Nup107 activity is continuously required in ECs for the formation of cellular extensions and regulation of GSCs differentiation” and new Figure 5 (line number 271).

5. Similarly, for the rescuing experiments, which are critical for drawing quite a few critical conclusions for this paper, proper control to avoid interpretation caveats should be included.

We agree and in our rescue experiments, we have provided the control with UAS-GFP. This is now clarified in figure legend 7 (line number 389). We therefore knocked-down *Nup107* and concomitantly overexpressed *Dsx^F^*, utilizing *GFP* overexpression as a control in the KD lines.

6. The reviewers also noticed quite some critical literatures regarding female gonadal development are missing.

We apologize for these inadvertent omissions and we have updated the list of references in the revised version. We have tried to include additional citations wherever necessary and will be happy to include more as per reviewer’s suggestions.

Reviewer #1 (Recommendations for the authors):Overall the studies were well done and the analyses are rigorous, here are some suggestions that will hopefully improve this study and make some conclusions more solid:General suggestions:1. For the cell type specificity for Nup107, I think in addition to the cell type-specific RNAi experiments, it would be more complete to add the cell type-specific rescuing experiments and results. For example, expression of Nup107 in Tj-positive somatic gonadal cells in the nup107 mutant background.

This is very complicated and cumbersome since we don’t have UAS-Nup107 flies, and in our mutant fly model, chromosomes 2 and 3 are occupied.

2. In several figures, nos-Gal4 driving RNAi does not yield to significant phenotype but tj-Gal4 does, I think it is important to show that the germline RNAi does lead to reduced levels of the target genes in germ cells.

As suggested by the reviewer we have now performed this experiment and have verified, using qRT-PCR, the efficiency of the germline KD using nos-Gal4 (Figure 1—figure supplement 1).

3. It has not been mentioned about the basic cellular mechanism of the functions of Nup107. For example, which cell type is Nup107 expressed and what is its subcellular localization? Does these patterns change in the mutant background? In the Discussion, it was mentioned that Nup107 is a "ubiquitously expressed nuclear envelope protein", but please show the results.

We explain in the introduction that “Nup107 is an essential component of the nuclear pore complex, enabling both active and passive transport in every nucleated cell” (line number 61).

In addition, in our previous paper we showed the ubiquitous expression of WT RFP-Nup107 and that the mutation does not affect this expression. (Weinberg-Shukron et al., 2015)

Specific suggestions on results and data analyses:4. Figure 1, Nup107 RNAi, need to quantify the KD efficiency. Also, in Figure 1G, what do the average and deviation stand for as the Y-axis is for the percentage, is it from a number of independent experiments? Also, in the figure legend, need to clarify what statistical method was used for the P values. Scale bars are also needed for panel B, C, D, E, and F.

As suggested, we have quantified the efficiency of the Nup107 KD in Figure 1—figure supplement 1. Average and deviation represent multiple independent experiments (>3). The statistical methods are detailed in the Materials and methods in the *Statistical Analysis* section (line number 697).

The scale bar in 1A applies to all of the panels which is now included in the legend (line number 114).

5. Regarding this statement "Thus, GSCs can be readily identified by the presence of a round fusome, whereas their differentiated daughter cells display extended, branched fusomes." In fact, both GSCs and CBs have a round spectrosome, while more differentiated cystocytes have branched fusomes.

We thank the reviewer for pointing this out and have corrected the manuscript as follow: “Thus, GSCs and their undifferentiated cystoblasts can be readily identified by the presence of a round fusome, whereas their differentiated daughter cells display extended, branched fusomes (line number 168).

6. Given both GSCs and CBs have spectrosomes and each ovariole contains 2-3 GSCs and probably around 2-3 CBs, I wonder whether for this statement: "Ovary staining revealed that unlike yw and Nup107WT germaria, which contained 2-3 spherical fusomes", the number would be a big larger than 2-3, and if this is true, then "an average of 6 spherical fusomes per germarium" may not be a phenotype?

Based on our observations, normal germaria contain <milestone-start />3<milestone-end />-<milestone-start />4<milestone-end /> spherical fusomes which include the GSCs and the CBs. Therefore, an average of 6 spherical fusomes per germarium is an aberrant phenotype. Our findings are in agreement with previous studies. See for example (Casanueva and Ferguson, 2004) who report “…we found that wild-type ovarioles have an average of 2.3±0.9 putative GSCs, and 1.2±0.8 putative CBs”. We are fully aware of other studies that report of 5.5 spectrosome containing cells per germarium (see for example: (Kirilly et al., 2011)). It is also possible that differences in diet or rearing conditions cause the observed differences.

7. Figure 3G quantifies germaria with few/no Cora extensions, as few is kind of subjective, it would be clearer to clarify the threshold for calling "few" in the figure legend. The same comments apply for Figure 5D quantification.

In the main text, we write: " In contrast, these cellular extensions were either dramatically reduced or completely lost". While we agree that the word "few" is subjective, in our case the majority of germaria are missing the extensions completely. Additionally, the comparison between those defined as having "few" and the amount in the controls is unequivocal: The difference between the two is obvious. Regardless, to resolve this issue, we have changed the figure legend to read: "Adult ovaries demonstrate missing EC extensions" (line number 241).

8. Please add more statistical analyses for panel 4D and 4J.

Figure 4 is renamed Figure 2. Performed as suggested (line number 145)

9. For RNA-seq, please show more analyses including biological replicates of the three different genotypes. Also, the authors mentioned that they used "LL3 larval gonads" for RNA-seq, I assume they mean larval ovaries but please specify it in the Results.

As per the reviewer insight, we have changed the wording to “LL3 female larval gonads” (line number 310). We have also clarified in the text that the RNA-Seq was performed in triplicates (line number 311).

10. Regarding the experiments to show "Dsx overexpression rescues the phenotypes induced by Nup107 loss", I think it is better to rescue driving Dsx_F_ using tj-Gal4 at the nup107 mutant background. The co-expression of both Dsx_F_ and nup107 RNAi using the same tj-Gal4 driver could result in a "rescue" due to competition for the same driver. Therefore, either rescue using the nup107 mutant background or for the rescue using tj> Dsx_F_ + nup107 RNAi, provide the control with tj> lacZ (or another non-specific UAS-transgene) + nup107 RNAi.

As mentioned above, this is very complicated since in our mutant fly model chromosomes 2 and 3 are occupied.

We have provided the control requested using UAS-GFP, which is now clarified in figure legend 7 (line number 389).

11. Regarding "AdamTS-A acts downstream of Nup107 and Dsx", does overexpression of AdamTS-A also rescue nup107 mutant phenotypes?

This was a very useful suggestion. We have now tested this and discovered that, unlike *dsx-F*, overexpression of AdamTS-A is insufficient to rescue the phenotypes induced by Nup107 (or even *Dsx*) loss. This result corroborates the central conclusion of our manuscript regarding *dsx-F* being a crucial downstream effector of Nup-107 function. This information has been included in the manuscript (line number 428).

Reviewer #2 (Recommendations for the authors):1. The authors perform somatic cell manipulation using tj-Gal4 for a large majority of their experiments and observe defects in 3-to-5-day old females. Therefore, defects that are observed in the ovary could be due to transgene expression during adulthood (which the authors acknowledge in lines 388-392) of the manuscript. A simple test would be to combine tj-Gal4 with tubPGal80ts (a temperature-sensitive allele of Gal80) and perform the Nup107 knockdown and determine whether there are ectopic GSCs similar as without the temporal control.

We thank the reviewer for this excellent suggestion. We have performed this experiment and have included the data in the revised version. Specifically, in order to identify the developmental stage where Nup107 activity is required, we performed temporal knockdown of Nup107 using a combination of *tj-Gal4* and a temperature sensitive form of the Gal4 repressor, Gal80ts: We allowed for normal ovarian development until hatching, at a non-permissive temperature, and only then knocked down Nup107 in ECs. Subsequent analysis of 4-5 days old female flies was performed. See new addition to manuscript, “Nup107 activity is continuously required in ECs for the formation of cellular extensions and regulation of GSCs differentiation” (line number 271).

2. It was previously shown that loss of Tkv in the developing soma results in ectopic GSCs in the adult ovary (Tseng et al., 2018), similar to results observed with loss of Nup107, dsx, and AdamTS-A (this study). Is BMP signaling in the developing gonad also disrupted?

This issue is very important that deserves a separate study. As such we believe that it is outside the scope of this particular manuscript however, we plan to address it in the future and thank the reviewer for this useful suggestion.

3. The authors propose that Nup107 is required for proper differentiation of escort cells. How is that determined when there were similar numbers of escort cells in tj-Gal4 > Nup107 RNAi compared to control. Loss of escort cell protrusions may not directly impact differentiation as there are many additional factors that can influence escort cell protrusions such as defects in EGFR signaling and changes in diet.

This was also an insightful suggestion. We have considered all these possibilities to the extent possible while revising the manuscript. We found that knockdown of Nup107 in ECs, with either tj-Gal4 or c587-Gal4, in fact, did not affect their visibility/presence compared to control (Figure 4A-D; line number 240, and Figure 4—figure supplement 1; line number 983), implying that the observed effects are not due to cell loss.

By knocking down Nup107 in ECs, we show that the resulting ECs are defective in their ability to form the cellular extensions. Importantly, we also performed new experiments, where we knocked down coracle expression using the c587-Gal4 driver, showing that indeed loss of ECs protrusions per se is sufficient to cause dysregulation of BMP signaling and impairment of germline differentiation (Figure 4—figure supplement 2; line numbers 993 and Figure 4—figure supplement 3 and line number 1004).

It should be noted that we did not manipulate the diet or compromised other signaling pathway such as EGFR. More studies will have to be conducted to assess if Nup107 also affects EGFR pathway directly.

4. The authors show images for cleaved caspase (Figure 6F-G), but there is not any explanation in the text. Are the authors proposing an additional requirement for Nup107 in older egg chambers?

We apologize for not explaining this adequately in the previous version. In fact, this experiment was conducted as a follow up to our previous published findings, which showed abnormal cell death in egg chambers upon Nup107 loss (Weinberg-Shukron et al., 2015). Here, we show that *dsx* overexpression is able to rescue that phenotype as well. We have now clarified this point in the revised version (line numbers 370-373).

5. The conclusion that AdamTS-A is a direct target of Nup107 and dsx would be greater strengthened by rescue analysis (overexpression of AdamTS-A in a tj-Gal4 > Nup107 or dsx RNAi).

This was an excellent suggestion. As recommended, we have now performed this experiment and found that overexpression of AdamTS-A is, in fact, insufficient to rescue the phenotypes induced by either Nup107 or *Dsx* loss, suggesting that other critical targets are also regulated by the Nup107-*Dsx* axis. We have incorporated these results into the manuscript (line numbers 428-435). Importantly, the negative outcome of this experiment establish that *dsx-F* is likely mediating effects of Nup-107 in this context and thus is a major mediator of this function. It is also consistent with the fact that both Nup-107 and *Dsx*-F share significant number of transcriptional targets and AdamTS-A is one such example.

Consistently, transcription factor *Dsx* has been previously shown to bind to AdamTS-A locus (Clough et al., 2014). As suggested by their findings, we found that loss of *Dsx* leads to a significant reduction in adamTS-A expression (Figure 8G). Taken together, these results indicate that *Dsx* regulates AdamTS-A directly. Overall, our results support a model where Nup107 regulates the expression of *Dsx*, either directly or indirectly, while *Dsx* directly regulates the transcription of multiple target genes including AdamTS-A. This important detail is now clarified in the revised manuscript (line number 586).

6. The authors should indicate how region 2a in the germarium is being determined.

We explain this in figure legend 3: “Region 2a is identified by the presence of 16-cell cystocytes which are adjoining the follicle stem cell border” (line number 210).

7. The 1B1 staining in Figure 2E is hard to see. Is there a better representative image that could be used?

Thanks for pointing this out. We have changed the picture as requested and the staining pattern is now more obvious in the revised figure, now Figure 3E.

8. It should be indicated throughout the text that BMP signaling and other defects observed in the adult germaria and ovary could be due to either developmental defects or adult-specific defects.

To avoid this, we have now performed stage dependent KD experiments which demonstrate that the observed BMP signaling defects in the adult ovaries are due to requirement of Nup107 activity at the adult stage. See new addition to manuscript, “Nup107 activity is continuously required in ECs for the formation of cellular extensions and regulation of GSCs differentiation” (line number 271) and new Figure 5.

9. What are "high" spherical fusomes (line 276)?

Thanks for pointing out the incorrect usage. We have changed this to “elevated number of spherical fusomes” (line number 339)

10. There isn't a "standard cornmeal yeast extract media" (see Lesperance and Broderick, 2020). Please provide the ingredients and recipe used for food. Also, please indicate whether standard food was supplemented with wet yeast paste.

We agree and have provided the information in the methods as suggested by the reviewer (line number 607).

Reviewer #3 (Recommendations for the authors):– Nup107 and Dsx mutants show degenerated ovary, and they show that this is likely due to differentiation block (with high pMad in many germ cells). However, as can be seen in Figure 2, the ovarioles seem to have normal-looking assembly lines of egg chamber, and I wonder what is the exact cause of ovary degeneration. Can you please add a bit more details of exactly how stem cell differentiation defects lead to ovary degeneration in Nup107 or Dsx depletion?

We appreciate this thoughtful remark by the referee. To clarify this point further, we have added the to the discussion the following explanation.

“… Together these data imply that Nup107 acts specifically in ICs enabling them to effectively interact with the PGCs. Consistent with this notion, loss of Nup107 affected the behavior of ICs such that these cells showed varying degrees of failure to mingle with the PGCs. A severe failure of ICs and PGCs to interact in the larval gonad is expected to cause an ovarian dysgenesis-like phenotype as it is essential for the germarium development and ovariole formation. Likewise, a milder failure of intermingling in the larval gonad would allow for the formation of adult ovaries. However, in the adult ovary Nup107 activity is further required in ECs for the formation of their cellular extensions and regulation of differentiation of the GSCs”

– One cannot call Dsx as a 'target' of Nup107. What is the mechanism by which Dsx expression is changed in nup107 mutant?

If we understand correctly reviewer is questioning the use of the term ‘target’ because we have not uncovered the mechanism underlying regulation of *dxx-F* transcription by Nup107. However, we employed the term target to indicate two related findings.

A); Nup107 is required for the regulation of *dsx* expression.

B): biological effects of Nup107 are mediated by *dsx-F* in large part as indicated by the efficient rescue of *Nup107* mutant phenotype by expression of *dxx-F,* supporting the claim that it is a functional target of Nup107.

But if the reviewer insists, we will be happy to rephrase and will refer to *dsx-F* as a ‘downstream mediator’ of Nup107.

– Is RNAi efficiency validated for all RNAi lines?

This was already done for several RNAi lines used in our study and as suggested by the reviewers, we have subsequently validated the efficacy of all the relevant RNAi lines using qRT-PCR assay (Figure 1—figure supplement 1), (line number 967).

– What I am most confused about is the apparent inconsistency of the Dsx mutant phenotype described in this study vs. previous studies (Erika Bach, Mark van Doren and Erika Matunis' lab). Perhaps the data are not inconsistent, but previous work focused on sex transformation, and I am puzzled why the ovary of Dsx mutant described here looks quite normal in terms of sex identity. And it would be extremely helpful if the authors can place their data in the context of the previous work.

This apparent discrepancy was pointed out by all the reviewers and we have dealt with this question in detail. Please refer to our answer to the second point of the editor.

References:

Camara, N., Whitworth, C., Dove, A., and Van Doren, M. (2019). Doublesex controls specification and maintenance of the gonad stem cell niches in *Drosophila*. Development *146*, dev170001. 10.1242/dev.170001.

Casanueva, M.O., and Ferguson, E.L. (2004). Germline stem cell number in the *Drosophila* ovary is regulated by redundant mechanisms that control Dpp signaling. Development *131*, 1881-1890. 10.1242/dev.01076.

Clough, E., Jimenez, E., Kim, Y.A., Whitworth, C., Neville, M.C., Hempel, L.U., Pavlou, H.J., Chen, Z.X., Sturgill, D., Dale, R.K., et al. (2014). Sex- and tissue-specific functions of *Drosophila* doublesex transcription factor target genes. Dev Cell *31*, 761-773. 10.1016/j.devcel.2014.11.021.

Grmai, L., Hudry, B., Miguel-Aliaga, I., and Bach, E.A. (2018). Chinmo prevents transformer alternative splicing to maintain male sex identity. PLoS Genet *14*, e1007203. 10.1371/journal.pgen.1007203.

Kirilly, D., Wang, S., and Xie, T. (2011). Self-maintained escort cells form a germline stem cell differentiation niche. Development *138*, 5087-5097. 10.1242/dev.067850.

Ma, Q., Wawersik, M., and Matunis, E.L. (2014). The Jak-STAT target Chinmo prevents sex transformation of adult stem cells in the *Drosophila* testis niche. Developmental cell *31*, 474-486. 10.1016/j.devcel.2014.10.004.

Tseng, C.Y., Su, Y.H., Yang, S.M., Lin, K.Y., Lai, C.M., Rastegari, E., Amartuvshin, O., Cho, Y., Cai, Y., and Hsu, H.J. (2018). Smad-Independent BMP Signaling in Somatic Cells Limits the Size of the Germline Stem Cell Pool. Stem Cell Reports *11*, 811-827. 10.1016/j.stemcr.2018.07.008.

Weinberg-Shukron, A., Renbaum, P., Kalifa, R., Zeligson, S., Ben-Neriah, Z., Dreifuss, A., Abu-Rayyan, A., Maatuk, N., Fardian, N., Rekler, D., et al. (2015). A mutation in the nucleoporin-107 gene causes XX gonadal dysgenesis. J Clin Invest *125*, 4295-4304. 10.1172/JCI83553.